# Global biome changes over the last 21,000 years inferred from model-data comparisons

Chenzhi Li[1,2], Anne Dallmeyer[3], Jian Ni[4], Manuel Chevalier[5], Matteo Willeit[6], Andrei A. Andreev[1], Xianyong Cao[7], Laura Schild[1,2], Birgit Heim[1], Mareike Wieczorek[1], Ulrike Herzschuh[1,2,8]

[1] Polar Terrestrial Environmental Systems, Alfred Wegener Institute Helmholtz Centre for Polar and Marine Research, Telegrafenberg A45, 14473 Potsdam, Germany

[2] Institute of Environmental Science and Geography, University of Potsdam, Karl-Liebknecht-Straße 24-25, 14476 Potsdam, Germany

[3] Max Planck Institute for Meteorology, Bundesstraße 53, 20146 Hamburg, Germany

[4] College of Life Sciences, Zhejiang Normal University, 321004 Jinhua, China

[5] Institute of Geosciences, Sect. Meteorology, Rheinische Friedrich-Wilhelms-Universität Bonn, Auf dem Hügel 20, 53121 Bonn, Germany

[6] Department of Earth System Analysis, Potsdam Institute for Climate Impact Research, P.O. Box 60 12 03, 14412 Potsdam, Germany

[7] Alpine Paleoecology and Human Adaptation Group (ALPHA), State Key Laboratory of Tibetan Plateau Earth System, Environment and Resources (TPESER), Institute of Tibetan Plateau Research, Chinese Academy of Sciences, 100101 Beijing, China

[8] Institute of Biochemistry and Biology, University of Potsdam, Karl-Liebknecht-Straße 24-25, 14476 Potsdam, Germany

*Correspondence to:* Ulrike Herzschuh (Ulrike.Herzschuh@awi.de)

**Abstract.** We present a global megabiome reconstruction for 43 timeslices at 500-year intervals throughout the last 21,000 years,based on an updated and thus currently most extensive global taxonomically and temporally standardized fossil pollen dataset of 3,455 records. The evaluation with modern potential natural vegetation distributions yields an agreement of ~80%,suggesting a high reliability of the pollen-based megabiome reconstruction.

We compare the reconstruction with an ensemble of six biomized simulations derived from transient Earth System Models (ESMs). Overall, the global spatiotemporal patterns of megabiomes estimated by both, the simulation

ensemble and reconstructions, are generally consistent. Specifically, they reveal a global shift from open glacial non-forest megabiomes to Holocene forest megabiomes since the Last Glacial Maximum (LGM), in line with the general climate warming trend and continental ice-sheet retreat. The shift to a global megabiome distribution generally similar to today's took place during the early Holocene; Furthermore, the reconstructions reveal that enhanced anthropogenic disturbances since the Late Holocene have not altered broad-scale megabiome patterns.

However, certain data-model deviations are evident in specific regions and periods, which could be attributed to systematic climate biases in ESMs or biases in the pollen-based biomization method. For example, at a global scale over the last 21,000 years, the largest deviations between reconstructions and simulation ensemble are observed during the LGM and early deglaciation periods. These discrepancies are probably attributed to the ESM systematic summer cold biases that overestimate tundra in periglacial regions, as well as the challenging
identification of steppes and tundra from the Tibetan Plateau pollen records. Moderate deviations during the Holocene mainly occur in non-forest megabiomes in the Mediterranean and North Africa, with increasing discrepancies over time. These deviations may result from the underestimation of woody PFT cover in simulations due to systematic biases, such as overly warm summers with dry winters in the Mediterranean, and the overrepresentation of woody taxa in reconstructions, misclassifying deserts as savanna in North Africa.

Overall, our reconstruction, with its relatively high temporal and spatial resolution, serves as a robust dataset for evaluating ESM-based paleo-megabiome simulations, as well as providing potential clues for improving systematic model biases.

## 1 Introduction

Earth system models (ESMs) that incorporate vegetation dynamics are useful tools for understanding historical simulations and future projections of composition, structure, and distribution changes of vegetation ecosystems, as well as their responses and feedbacks to climate change (Song et al., 2021; Brierley et al., 2020; Song et al., 2021). However, to assess model biases and further improve these models for obtaining more reliable and reduced uncertainty in future projections, global and long-term paleo-vegetation reconstructions are needed for the
evaluation of the vegetation response to climate change (Cao et al., 2019; Dallmeyer et al., 2022). Pollen records, as the most widespread terrestrial paleoecological archives, and their conversion into paleo-vegetation are most suitable for this purpose (Prentice et al., 1996). To date, however, the synthesis of global-scale pollen-based vegetation reconstructions has been limited to selected timeslices (i.e., mid-Holocene and LGM); Harrison, 2017; Hoogakker et al., 2016; Harrison, 2017), while continuous reconstructions have been limited to specific regions
(such as Northern and Eastern Asia, extratropical Northern Hemisphere; Tian et al., 2018; Cao et al., 2019). A global view of reconstructed vegetation dynamics and distributions since the LGM with high temporal resolution is still missing.

In a recent effort, we synthesized LegacyPollen 2.0 (Li et al., 2025), a taxonomically and temporally standardized global Late Quaternary fossil pollen dataset of 3,680 records that covers the main global ecoregions (Herzschuh
et al., 2022). In this study, we biomize the LegacyPollen 2.0 dataset for 43 timeslices at 500-year intervals throughout the last 21,000 years with a biomization method (Prentice et al., 1996; Prentice and Webb, 1998) that

incorporates updated and harmonized pollen taxa-plant functional types-megabiome assignment schemes. For a direct comparison with ESM-simulated vegetation, we assign the reconstructions into the same megabiomes used in the biomization tool for ESM output by Dallmeyer et al. (2019). This paper aims (a) to present megabiome dynamics at the global scale since the LGM, (b) to compare the reconstruction with megabiome simulations from an ensemble of six different transient ESM simulations, and (c) to identify regions and periods with strong data-model mismatches to provide clues for improving systematic model biases.

## 2. Data and methods

### 2.1 Pollen dataset

We expanded the LegacyPollen 1.0 dataset (Herzschuh et al., 2021, 2022) to LegacyPollen 2.0, a taxonomically and temporally standardized global late Quaternary fossil pollen dataset (Fig. A1). The updated dataset comprises 3,680 palynological records, approximately 900 more than the previous LegacyPollen 1.0 dataset. Of these new records, 654 were derived from the Neotoma Paleoecology Database (Neotoma hereafter; https://www.neotomadb.org/; Williams et al., 2018) and its constituent databases (e.g., African Pollen Database (APD; Lézine et al., 2021), European Pollen Database and Alpine Pollen Database (EPD and ALPADABA; Fyfe et al., 2009; Giesecke et al., 2014), and Latin American Pollen Database (LAPD; Flantua et al., 2015). Also, 52 records from the Abrupt Climate change and Environmental Responses (ACER) 1.0 database (Sánchez Goñi et al., 2017a,b), 177 records from the Chinese fossil pollen dataset (Cao et al., 2022; Zhou et al., 2023), and 8 of our own new records (AWI; for a detailed description see Supplementary Data 1) were included. A total of 1122 records originate from North America, 1446 from Europe, 687 from Asia, 187 from South America, 159 from Africa, and 81 from the Indo-Pacific. While there are geographical gaps in pollen record coverage, particularly in the Southern Hemisphere, the dataset LegacyPollen 2.0 covers the world's main vegetation and climate zones.

To improve comparability between pollen records as well as data quality, we followed the practices recommended by Flantua et al. (2023) for large-scale paleoecological data synthesis when updating the dataset. Specifically, the following key steps were involved: first, metadata of pollen records from different data sources were examined to avoid duplicate inclusion; second, age-depth models were re-estimated for each record ($\geq$ 2 radiocarbon dates) using Bacon (Blaauw and Christen, 2011; for a detailed description, see Li et al., 2022); third, pollen morphotypes were harmonized to reduce the effect of taxonomic uncertainty and nomenclatural complexity, i.e. woody taxa and major herbaceous taxa have been harmonized to genus level and other herbaceous taxa to family level (for a detailed description see Herzschuh et al., 2022).

The LegacyPollen 2.0 dataset is archived in the PANGAEA repository (https://doi.org/10.1594/PANGAEA.965907; Li et al., 2025) and is open-access. It follows the framework of LegacyPollen 1.0 dataset (Herzschuh et al., 2021, 2022), providing pollen count and pollen percentage data per continent, a taxa harmonization master table, and site metadata (such as data sources, Dataset ID, site name, location, archive type, site description, and references). To enhance data traceability and ensure high-quality standards, we have newly incorporated the Neotoma digital object identifier (DOI) into the metadata for Neotoma-derived records, allowing direct linkage to the living Neotoma database and reducing the risk of data staleness.

These DOIs were generated with the *doi*() function from the package *neotoma2* (version 1.0.3; Socorro and Goring, 2024) in the R software environment (version 4.4.1; R Core Team, 2023). Additionally, we also newly added the PANGAEA Event (PANGAEA dataset identifier) for each record to ensure that our dataset meets PANGAEA's high standards for quality, usability, and compliance.

**2.2 Pollen-based megabiome reconstruction**

We converted pollen data from LegacyPollen 2.0 into megabiomes using the biomization method of Prentice et al. (1996). We only analyzed records over the last 21,000 years, resulting in a final megabiome dataset of 55,868 samples at 500-year intervals from 3,455 records (Supplementary Data 1 and Data 4). The assignment of pollen taxa to plant functional types (PFTs), the first step required by the biomization procedure, referenced previous biomization schemes on each continent, with some updates and harmonizations based on a globally applicable standardized classification of PFTs (Harrison et al. 2010; Harrison, 2017). The PFTs were then assigned to megabiomes, representing the raw pattern of global vegetation rather than the finer biome categories commonly used in standard biomization studies (Dallmeyer et al., 2019). These megabiomes include tropical forest (TRFO), subtropical forest (WTFO), temperate forest (TEFO), boreal forest (BOFO), (warm-) savanna and dry woodland (SAVA), grassland and dry shrubland (STEP), (warm) desert (DESE), tundra and polar desert (TUND). These categories were also applied to biomize Earth System Model results, which generally use different types and numbers of PFTs to represent global vegetation, enabling direct data-model comparisons and evaluations (Dallmeyer et al., 2019). The pollen sample at each target timeslice was selected from the time-nearest sample within ± 250 years.

We assigned the 1447 harmonized pollen taxa from the 3,455 records to 98 PFTs and then to 8 megabiomes. The pollen abundances of *Larix* and *Pinus* were multiplied by factors of 15 and 0.5 (following Bigelow et al., 2003 and Cao et al., 2019), respectively, to compensate to some extent for pollen productivity-related representativeness issues, prior to calculating affinity scores in the applied biomization routine. When the affinity scores for each megabiome were calculated (cf. Prentice et al., 1996) for every pollen sample, pollen taxa with less than 0.5% abundance were excluded to reduce noise resulting from occasional pollen grains derived from long-distance transport or contamination (Prentice et al., 1996; Chen et al., 2010). Finally, the megabiome with the highest affinity score was allocated to each pollen sample, subject to a criterion that the least PFT-rich megabiome takes precedence when the affinity values for two or more megabiomes are identical (following Chen et al., 2010). The biomization affinity scores were calculated using a biomization algorithm implemented in R (Cao and Tian, 2021). Furthermore, the assignment of pollen taxa to megabiomes and biomization routines were performed independently for each continent (Table 1; Supplementary Data 2 and Data 3).

**Table 1. Overview of the number of pollen records, pollen taxa, plant functional types (PFTs), and megabiomes used in the biomization procedures, along with references to used biomization schemes by continents.** The lists of taxa-PFTs and PFTs-megabiome assignments are available in Supplementary Data 2 and Data 3.

| Continent | Pollen records | Taxa | PFTs | Megabiomes | References |
|---|---|---|---|---|---|
| Europe | 1,359 | 243 | 41 | 7 | Ni et al. (2014) |
| | | | | | Binney et al. (2017) |
| | | | | | Marinova et al. (2018) |
| | | | | | Cao et al. (2019) |
| Asia | 636 | 424 | 49 | 8 | Chen et al. (2010) |
| | | | | | Ni et al. (2014) |
| | | | | | Binney et al. (2017) |
| | | | | | Tian et al. (2018) |
| | | | | | Cao et al. (2019) |
| North America | 1,078 | 393 | 47 | 8 | Thompson and Anderson (2000) |
| | | | | | Ortega-Rosas et al. (2008) |
| | | | | | Bigelow et al. (2003) |
| | | | | | Ni et al. (2014) |
| | | | | | Cao et al. (2019) |
| Africa | 145 | 556 | 8 | 6 | Vincens et al. (2006) |
| | | | | | Lézine et al. (2009) |
| Indo-Pacific | 60 | 429 | 22 | 8 | Pickett et al. (2004) |
| South America | 177 | 576 | 19 | 8 | Marchant et al. (2001 & 2009) |
| Total | 3,455 | 1,447 | 98 | 8 | |

**2.3 Transient ESM-based simulations with dynamic vegetation**

We use six transient simulations for the last 21,000 years performed with Earth System Models with fully coupled dynamic vegetation. Among these are two simulations conducted with the Max-Planck-Institute Earth-System-Model (MPI-ESM; Mauritsen et al., 2019), further referred to as MPI-ESM_GLAC1D (Dallmeyer et al., 2022) and MPI-ESM_ICE6G (Ice6G_P2 in Kapsch et al., 2022; Mikolajewicz et al., 2023). Besides differences in the model version and tuning, these simulations differ in particular with respect to the prescribed ice-sheet history, using either the GLAC-1D (Tarasov et al., 2012) or ICE-6G (Peltier et al., 2015) reconstruction. Both simulations ran at the spatial resolution T31 (~3.75°x3.75° on a Gaussian grid) for the atmosphere and land model. Orbital forcing has been prescribed from Berger (1978) and greenhouse gas (GHG) forcings from Köhler et al. (2017). Bathymetry, topography, and river routing were continuously updated in ten-year intervals. The meltwater flux from the Laurentide ice sheet has been modified in the period of 15.2–11.8 cal. ka BP (calibrated thousand years before present, where "present" is 1950 C.E.) in the simulation MPI-ESM-GLAC1D, mimicking the meltwater storage and release from proglacial lakes and thus more realistically simulate the Younger Dryas event (cf. Dallmeyer et al., 2022).

In addition, the set of simulations includes the full-forcing TRACE-21K-I (cf. Liu et al., 2009) and TRACE-21K-II (cf. He and Clark, 2022) simulations performed with the Community Climate Model version 3 (CCSM3, Collins et al., 2006) forced with variations in insolation (Berger, 1978), GHG concentration (Joos and Spahni, 2008), and continental ice sheets from the ICE5G reconstructions (Peltier, 2004). TRACE-21K-II was based on the protocol of prescribing the reconstructed Atlantic meridional overturning circulation (AMOC) for the Bølling-Allerød interstadial (~14.7–12.9 cal. ka BP) and the Holocene instead of the reconstructed freshwater forcing, while in

TRACE-21K-I, the AMOC has been forced by the meltwater flux to the North Atlantic and the Gulf of Mexico during the entire simulation. Similar to the MPI-ESM simulations, the TRACE-21K simulations ran at a spatial resolution of T31 (~3.75°x3.75° on a Gaussian grid).

The set of simulations contains two simulations performed with the fast Earth System model CLIMBER-X (Willeit and Ganopolski, 2016; Willeit et al., 2022, 2023) at a spatial resolution of 5°x5°. These simulations were both performed in an identical setup (similar to Masoum et al., 2024) but with different ice-sheet and surface topography forcings (GLAC-1D or ICE-6G reconstructions; Peltier et al., 2015; Tarasov et al., 2012). GHG and insolation have been prescribed from Köhler et al. (2017) and Laskar et al. (2004), respectively.

All paleoclimate simulations have been aggregated to time series of 100-year monthly climatological means. The first timeslice at 21 cal. ka BP is an average of the years 21,099–21,000 years before present (where "present" is 1950 C.E.), and the last timeslice at 0 cal. ka BP is an average of the years 99–0 years before present.

The dynamic vegetation in all models is represented by different sets of plant functional types (PFTs) that can coexist in the grid-cells. The occurrence of each PFT is constrained by fixed temperature thresholds, and the dynamics of PFT cover fraction depends for instance on the moisture availability and plant requirements. Disturbances such as fire, which are already coupled in the dynamic vegetation modules, regularly reduce the coverage of tree and shrub PFTs while promoting the expansion of herbaceous PFTs (Burton et al., 2019; Reick et al., 2021; Dallmeyer et al., 2022). Land use is not included in any of these simulations.

The PFTs distributions are converted into the same eight megabiomes used in the reconstructions by applying the tool of Dallmeyer et al. (2019). This tool converts the simulated PFT distributions based on assumptions of the minimum PFT cover fractions that are needed for the assignment of steppe/tundra or forest biomes and bioclimatic constraints derived from 2 m surface temperature distributions to distinguish different forest biomes (for a detailed description see Dallmeyer et al., 2019). These constraints largely adhere to the limitation rules used in the classical biome models such as BIOME4 (Kaplan et al., 2003).

We assigned the simulated megabiome data taken from the grid-cells where the records are located to each record, and we only considered locations and timeslices for which reconstructions are available (Supplementary Data 4 - 6). As representatives of the simulation ensemble, we choose the megabiome that occurs most frequently in the set of simulations for each record and timeslice, further referred to as the ESM-representative megabiome. When the highest-frequency megabiomes were not unique, we applied the criterion used in pollen-based reconstructions, giving precedence to the highest-frequency megabiome with the fewest PFTs and taxa.

**2.4 Evaluation with modern climate and potential natural vegetation**

Modern observational climate data provide a crucial foundation for the assessment of climate simulations. The Climatic Research Unit gridded Time Series (CRU TS hereafter), version 4.08, is a widely used modern observational climate dataset covering all land domains of the world except Antarctica (spatial resolution: ~0.5°x0.5° on a Gaussian grid; Harris et al., 2020). The CRU TS dataset is interpolated from extensive networks of weather station observations and provides monthly temperature and precipitation data from 1901-2023 C.E. However, the early records (i.e., < 1930 C.E.) of this dataset may have high uncertainty due to sparser observation

networks (Duan et al., 2024), and the late records (i.e., > 1970 C.E.) is strongly influenced by anthropogenic $CO_2$ increases (Cheng et al., 2022). We, therefore, selected monthly climatological means from 1931-1970 to generate more biologically meaningful bioclimatic variables for evaluating climate simulations at 0 cal. ka BP (O'Donnell and Ignizio, 2012; Supplementary Data 7). These bioclimatic variables represent extreme or limiting environmental factors, namely, mean temperature of warmest quarter ($T_{warm}$), mean temperature of coldest quarter ($T_{cold}$), precipitation of warmest quarter ($P_{warm}$), and precipitation of coldest quarter ($P_{cold}$). Temperature is given in degrees Celsius (°C), precipitation in millimeters (mm), and a quarter is a period of three consecutive months (1/4 of the year).

Modern vegetation distributions are required to validate the performance of pollen-based megabiome reconstructions and ESM-based megabiome simulations. However, the simulations used here only determine potential natural vegetation in a quasi-equilibrium with climate, whereas the pollen-based reconstruction of modern vegetation also incorporates anthropogenic disturbances. Therefore, the modern potential natural vegetation distributions are used for validation, allowing us to evaluate not only the level of modern anthropogenic disturbance to natural vegetation in the pollen-based reconstructions but also simulation biases. For this purpose, we employed the modern potential natural vegetation distribution (spatial resolution: 5 arc minutes) provided by Ramankutty et al. (2010). It represents the world's vegetation cover that would have most likely existed for 1986–1995 C.E. in equilibrium with present-day climate and natural disturbances in the absence of human activities (Ramankutty and Foley, 1999). To allow direct comparisons between reconstructions and simulations, as well as among simulations at the hemispheric or continental scales, we aggregated the modern potential natural vegetation types into modern potential megabiomes (Fig. 1) following Dallmeyer et al. (2019).

To assess the accuracy of the pollen-based reconstructions and ESM-based simulations, we calculated the proportion of records where reconstructed or simulated megabiomes at timeslice 0 cal. ka BP match these modern potential megabiomes. For each record, the simulated (most-representative) megabiome at timeslice 0 cal. ka BP and the modern potential megabiome were extracted from the grid-cells in which the record is located.

**2.5 Methods for comparison of the simulated and reconstructed megabiome datasets**

The Earth mover's distance (EMD), which takes into account the uncertainties of the biomized data and case-specific weighted distances (Chevalier et al., 2023b), was applied to quantify the mismatch between the pollen-based reconstructions and ESM-based simulation ensemble at each record. Specifically, the EMD calculates the distance between the reconstruction and simulation ensemble by considering the entire range of megabiome affinity scores. This means that the details of the underlying vegetation structure are part of the comparison, in contrast to commonly used methods that solely compare the megabiome with the highest affinity score estimated from the reconstructions or simulations. To match the distribution of megabiome scores obtained from biomization algorithms, we translated the frequencies of the six simulated megabiomes into a simulated megabiome affinity score set. For example, for an ensemble of simulations with two boreal forests and four temperate forests in its six simulations, the affinity scores for the boreal and temperate forests would be 2/6 and 4/6, respectively, while the affinity scores for the remaining megabiomes would be zero. In addition, we adapted the ecological and climatic distance-based (Allen et al., 2020) EMD weighting scheme from Chevalier et al. (2023b) to penalize mismatches between the reconstructions and simulation ensemble in terms of differences in vegetation structure (i.e., forest

megabiomes, non-forest megabiomes, and deserts) and climate zone preferences (i.e., tropical, subtropical, temperate, boreal, and polar regions) (Table 2). Following this approach, we assume that the basal distance between two different megabiomes with the same vegetation structure and climate zone is set to 1. Each difference in vegetation structure or climate zone adds an extra weight of 1. For example, the reconstructed tropical forest has a distance weight of two from the simulated temperate forest and three from the simulated boreal forest. The EMD routines were implemented by using the *paleotools* R package (version 0.1.0; Chevalier, 2023a).

**Table 2. Earth mover's distance (EMD) weighting scheme for ecological and climatic distances between the pollen-based reconstructed and simulated megabiomes used in this study.** Higher values in the table indicate a greater ecological ecological or climatic distance between the reconstructed and simulated megabiomes. Megabiome code: TRFO- tropical forest, WTFO- subtropical forest, TEFO- temperate forest, BOFO- boreal forest, SAVA- (warm) savanna and dry woodland, STEP-grassland and dry shrubland, DESE- (warm) desert, TUND- tundra and polar desert. Of these, TRFO, WTFO, TEFO, and BOFO are forest megabiomes, whereas the others are non-forest megabiomes.

| Reconstruction vs. Simulation | TRFO | WTFO | TEFO | BOFO | SAVA | STEP | DESE | TUND |
|---|---|---|---|---|---|---|---|---|
| **TRFO** | 0 | 1 | 2 | 3 | 1 | 2 | 3 | 4 |
| **WTFO** | 1 | 0 | 1 | 2 | 2 | 2 | 3 | 3 |
| **TEFO** | 2 | 1 | 0 | 1 | 3 | 2 | 3 | 2 |
| **BOFO** | 3 | 2 | 1 | 0 | 4 | 3 | 2 | 1 |
| **SAVA** | 1 | 2 | 3 | 4 | 0 | 1 | 2 | 4 |
| **STEP** | 2 | 2 | 2 | 3 | 1 | 0 | 1 | 1 |
| **DESE** | 3 | 3 | 3 | 2 | 2 | 1 | 0 | 1 |
| **TUND** | 4 | 3 | 2 | 1 | 4 | 1 | 1 | 0 |

We aggregated the records into regular longitude-latitude grid-cells of size 3.75°x3.75° to reduce the sampling bias from the non-uniform spatial distribution of records and to facilitate a more direct model-data comparison. At each timeslice, the reconstructed or simulated megabiome assigned to a grid-cell was determined based on the most frequently occurring megabiome among the available records in that grid-cell. When multiple megabiomes had the same highest frequency, we applied the same criterion used in pollen-based reconstructions, prioritizing the highest-frequency megabiome with the fewest PFTs and taxa. Similarly, the data-model EMDs for each grid-cell were derived as the median EMDs of the available records within that grid-cell.

To cluster the regions, we performed the dynamic time warping with the time series of the data-model EMDs of all grid-cells on each continent, which allows time series to be grouped based on their patterns or shapes (Müller et al., 2007). The number of clusters was determined using the elbow method (Syakur et al., 2018) and adjusted based on the sample availability. The global data-model EMDs time series, representing the global mean dynamics, was then synthesized from the median EMDs for each clustered region. The dynamic time warping algorithm was implemented by using the *TSclust* R package (version 1.3.1; Montero and Vilar, 2015).

# 3. Results and discussion

## 3.1 Evaluation of megabiome reconstructions and simulations for the present-day

### 3.1.1 Pollen-based reconstructions

We consider global-scale, pollen-based megabiome reconstructions to be reliable, as record-by-record comparisons of reconstructed megabiomes at timeslice 0 cal. ka BP from 2,232 available records with modern potential megabiomes indicate an 80.2% agreement (Table 3). This consistency exceeds that reported in previous large-scale biomization studies validated against modern biome distributions, such as the 53% agreement in Arctic high-latitudes (>55°N) by Bigelow et al. (2003). We attribute this high agreement not only to the high quality of the pollen dataset, particularly in terms of taxonomic and temporal harmonization, but also to the biomization method that employs updated and harmonized schemes assigning pollen taxa to plant functional types to megabiomes. Additionally, our reconstruction was conducted at the megabiome level, a coarser classification than typical biomes, which somewhat reduces mismatches between geographically adjacent biomes. For instance, the biomes of temperate deciduous forest and cool mixed forest are often intermingled in Binney et al. (2017), whereas at the megabiome level, both are classified as temperate forests, eliminating this discrepancy. Although some regional-scale biomization studies achieve even higher agreement with modern biome distributions, such as the 97.5% accuracy in the Congo Basin reported by Lebamba et al. (2009), these studies typically rely on more localized datasets with tailored taxa-PFT-biome schemes. Moreover, not all megabiomes were reconstructed with the same level of accuracy in our study, such as the TUND and STEP exhibit only ~50% agreement (Table 3), which similar to previous biomization studies. Overall, we argue that the data quality as well as the higher spatial and temporal coverage compared to previous biomization studies (Bigelow et al., 2003; Marinova et al., 2018) make our pollen-based megabiome reconstruction a robust dataset for various applications, such as global-scale evaluation of paleo-simulations from Earth System Models (ESMs).

**Table 3. Agreement of modern potential megabiomes, aggregated from modern potential natural vegetation, with (a) pollen-based reconstructions and (b-h) simulations at 0 cal. ka BP.** We use a set of six transient simulations that have been run in an Earth System Model: **(c-d)** MPI-ESM (MPI-ESM_GLAC1D, MPI-ESM_ICE6G), **(e-f)** CLIMBER-X (CLIMBER-X_GLAC1D, CLIMBER-X_ICE6G), and **(g-h)** CCSM3 (TRACE-21K-I_ICE5G, TRACE-21K-II_ICE5G), as well as **(b)** ESM-representative megabiome that occur most frequently in the set of simulations. The megabiome codes are given in Table 2.

| | Record number at 0 ka | (a) Pollen-based reconstruction | (b) ESM-representative Simulation | MPI-ESM | | CLIMBER-X | | CCSM3 | |
|---|---|---|---|---|---|---|---|---|---|
| | | | | (c) MPI-ESM_GLAC1D | (d) MPI-ESM_ICE6G | (e) CLIMBER-X_GLAC1D | (f) CLIMBER-X_ICE6G | (g) TRACE-21K-I_ICE5G | (h) TRACE-21K-II_ICE5G |
| TRFO | 112 | 81.2 % | 86.6 % | 71.4 % | 62.5 % | 81.2 % | 69.6 % | 62.5 % | 56.2 % |
| WTFO | 59 | 78.0 % | 49.2 % | 35.6 % | 10.2 % | 50.8 % | 49.2 % | 42.4 % | 42.4 % |
| TEFO | 1249 | 86.9 % | 74.0 % | 77.0 % | 75.0 % | 66.0 % | 49.3 % | 11.2 % | 15.5 % |
| BOFO | 464 | 79.1 % | 52.4 % | 40.5 % | 35.3 % | 49.6 % | 38.4 % | 14.7 % | 36.0 % |
| SAVA | 57 | 77.2 % | 7.0 % | 3.5 % | 29.8 % | 1.8 % | 0.0 % | 0.0 % | 1.8 % |
| STEP | 163 | 52.8 % | 45.4 % | 20.9 % | 33.7 % | 38.0 % | 40.5 % | 43.6 % | 38.0 % |
| DESE | 22 | 72.7 % | 50.0 % | 59.1 % | 45.5 % | 18.2 % | 40.9 % | 45.5 % | 50.0 % |
| TUND | 106 | 50.0 % | 46.2 % | 40.6 % | 29.2 % | 28.3 % | 33.3 % | 59.4 % | 55.7 % |
| Overall | 2232 | 80.2 % | 64.1 % | 60.2 % | 57.8 % | 57.0 % | 45.3 % | 20.0 % | 26.1 % |

Several factors may contribute to the incorrect reconstruction of modern potential megabiomes in our study (Fig. 1). (a) The different pollen representation (including production, dispersion, and preservation) of plant taxa is the principal reason for inadequate separation of forest and open landscape ecotones. For example, the high pollen productivity of key taxa (such as *Artemisia*; Xu et al., 2014) results in an overestimation of grasslands and dry shrublands (STEP) in the East Asian summer monsoon northern marginal zone and the Great Plains of North America. Studies on pollen productivity and dispersal ability to date are mostly limited to a few taxa in north-central Europe and China (Wieczorek and Herzschuh, 2020), which limits large-scale calibration of pollen representation. (b) The low taxonomic resolution could also cause mismatches between neighboring forest megabiomes, as well as between tundra (TUND) and grassland (STEP). Woody taxa have been harmonized to the genus level rather than the species level, while herbaceous taxa are generally harmonized to the family level, except for common taxa like *Artemisia*, *Thalictrum*, and *Rumex*. This reduces the ecological information available for PFT assignment (Chen et al., 2010). For instance, different species within *Pinus*, *Alnus*, *Fagus*, and *Betula* (Tian et al., 2018) have different bioclimatic controls, phenology, and life forms, but identification at the genus level results in them being shared by key PFTs in different forest megabiomes (e.g., WTFO vs. TEFO, TEFO vs. BOFO) when assigning taxa to PFTs. One of the typical areas in which this problem occurs is southern Scandinavia. Pollen grains from *Betula pendula* in temperate forests and *Betula pubescens* in boreal forests (Beck et al., 2016) in this region can only be identified to genus level, resulting in these two key species not being good indicators of temperate and boreal forests. Similarly, TUND may have been misrepresented as STEP on the Tibetan Plateau. This misrepresentation can be attributed to their share of dominant characteristic taxa within Poaceae and Cyperaceae. However, STEP is defined by fewer PFTs and therefore preferentially allocated to samples. In contrast, woody PFTs are generally not defined in STEP, leading to a potential misallocation to TUND rather than STEP in cases of woody pollen grain occurrences (from long-distance transportation or local existence) in open landscape samples (Marinova et al., 2018; Chen et al., 2010), such as mismatches in southern Europe. (c) Anthropogenic modification of pollen assemblages has, to some extent, contributed to mismatches in forested areas. For example, incorrectly reconstructed grasslands and dry shrublands (STEP) in Northern China may reflect intensive land use (e.g., deforestation). However, the modern anthropogenic megabiomes are not well reconstructed at a broad spatial scale here, as with previous studies (Ni et al., 2014; Cao et al., 2022). We suggest that this may be related to the absence of anthropogenic PFTs and megabiomes in our taxa-PFT-megabiome assignment schemes, as well as the difficulty of distinguishing between anthropogenic and non-anthropogenic pollen when using genus or family levels (e.g., Poaceae, Rosaceae), and pollen samples generally being collected from records with less human disturbance.

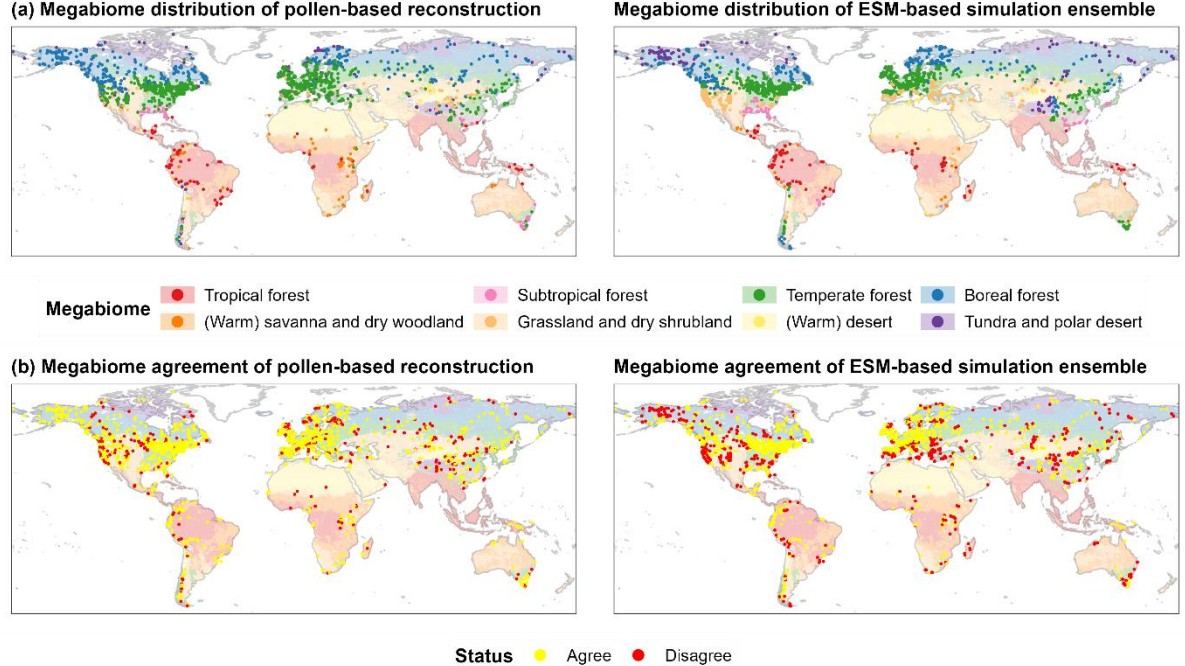

**Figure 1. Spatial patterns of megabiome distributions at 0 cal. ka BP (upper) and their agreement with modern potential natural megabiomes (lower), for each record derived from the pollen-based reconstruction and the ESM-based simulation ensemble.** Shown here are the ESM-representative megabiomes that occur most frequently in the set of simulations. The background depicts modern potential megabiomes (Dallmeyer et al., 2019) aggregated from modern potential natural vegetation (spatial resolution: 5 arc minutes; Ramankutty and Foley, 1999; Ramankutty et al., 2010), representing the world's vegetation cover that had most likely existed for 1986–1995 C.E. in equilibrium with present-day climate and natural disturbance in the absence of human activities.

### 3.1.2 ESM-based simulations

The agreement between modern potential megabiomes and simulated megabiomes at timeslice 0 cal. ka BP is higher for the ESM-representative megabiome (cf. Sect. 2.3) than for individual ESM-based simulations (64.1% vs. 20.0–60.2%; Table 3). As a result, the ESM-representative megabiome depicts more reliable patterns of megabiome dynamics and distribution than individual simulations, with higher agreement especially in Alaska, the Iberian Peninsula, the Alps, the Atlantic Coastal Plain of North America, and the southeastern United States (Fig. 1 and Fig. A2). However, there are still certain regions with low agreement, probably due to climatic biases. These include nearly all highlands (such as the central-southern Rocky Mountains, the central Andes, and the Tibetan Plateau) for which an overestimation of the temperature can be expected in the models due to a much lower orography than in reality caused by the smoothing in the coarse spatial resolution (3.75°x3.75° and 5°x5°) of the model grids (Fig. A3a–b). All models simulate non-forest megabiomes instead of forest in the Mediterranean region, which can be attributed to the models simulating a climate that is too seasonally dry, with, for example, too-warm summers and too-dry winters (Fig. A3a, d). The TRACE-21K simulation as well as the MPI-ESM simulations fail to reproduce the boreal forest (BOFO) in Alaska, which is then also reflected in the ESM-representative megabiomes. This failure is likely due to the simulated climate being too cold in this region, preventing the establishment of boreal forests under modeled conditions (Fig. A3a, d). Similar to the reconstructions, the transition zones between temperate forest (TEFO) and non-forest megabiomes, such as the

East Asian summer monsoon margin, are regions with lower simulated megabiome agreement to the modern potential megabiome distribution. In North Africa, the models also tend to underestimate the northern extension of the grassland and dry shrubland (STEP) and incorrectly assign (warm) savanna and dry woodland (SAVA) records to tropical forest (TRFO). This is related to the biomization procedure for the model results that only relies on simulated vegetation cover fractions and simulated climate, whereas savannas are additionally determined by other ecological processes such as fire intensity and frequency (Dallmeyer et al., 2019) or grazing (van Langevelde et al., 2019).

## 3.2 Global megabiome dynamics and distributions over the last 21,000 years

We present a global assessment of megabiome dynamics and distributions derived from pollen-based reconstructions and ESM-based simulations over the last 21,000 years, with a temporal resolution of 500 years. Overall, there has been a global shift from open glacial non-forest megabiomes to Holocene forest megabiomes since the LGM (Fig. 2), in line with the general climate warming trend and continental ice-sheet retreat (Fig. 3):

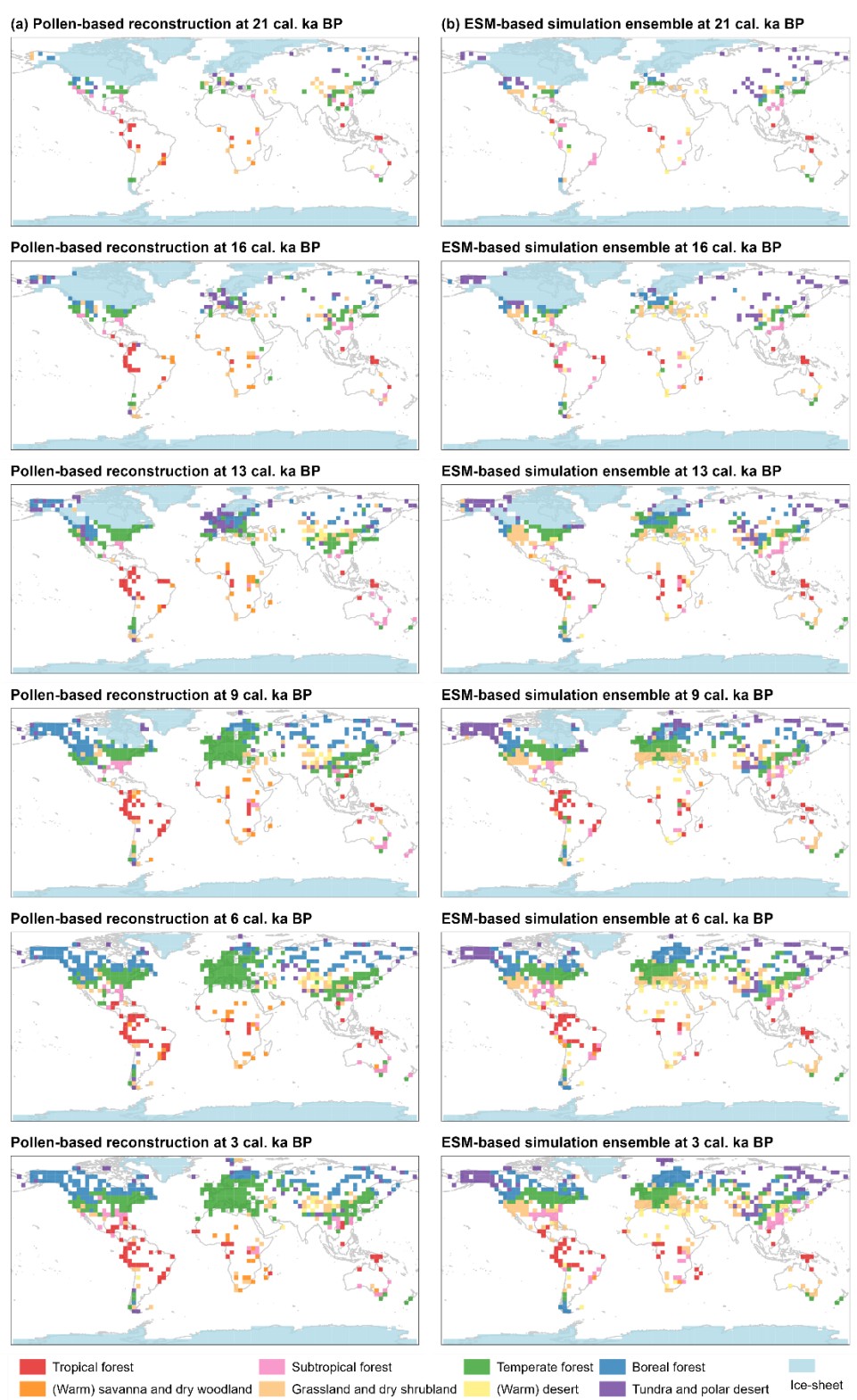

**Figure 2. Spatial distributions of megabiomes, derived from the (a) pollen-based reconstruction and (b) the ESM-based simulation ensemble, as well as the ice-sheet ensemble, at 21, 16, 13, 9, 6, and 3 cal. ka BP based on grid-cells of 3.75°x3.75°.** Shown here are the ESM-representative megabiomes that occur most frequently in the set of simulations. The ice sheets are shown at their maximum extent at timeslices synthesized for the ICE-5G (Peltier, 2004), ICE-6G (Peltier et al., 2015), and GLAC-1D (Tarasov et al., 2012) reconstructions.

360

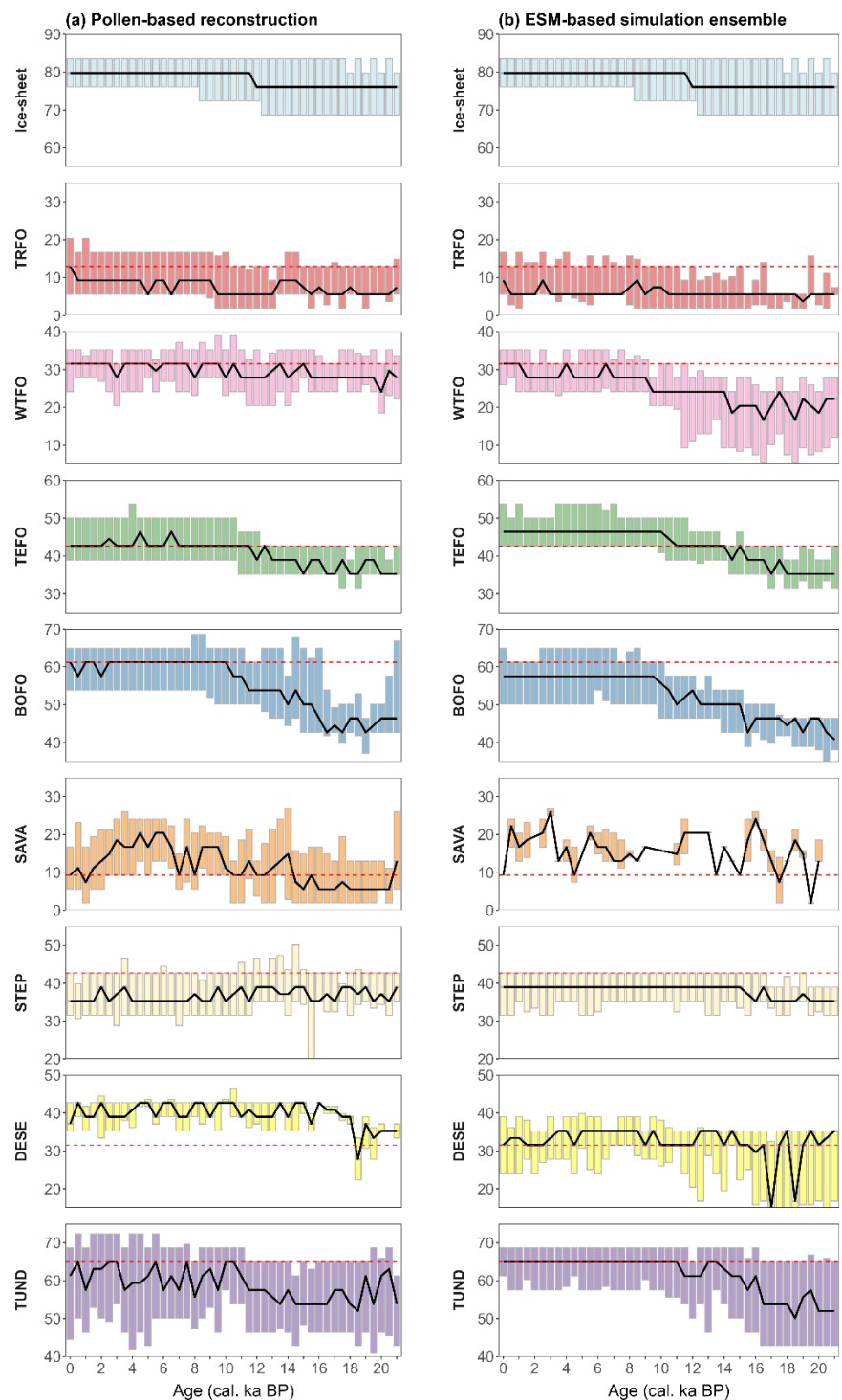

**Figure 3. Temporal changes in the latitudinal location (°) of each megabiome, derived from the (a) pollen-based reconstruction and (b) ESM-based simulation ensemble, as well as the ice-sheet ensemble, based on grid-cells of 3.75°x3.75° over the last 21,000 years globally.** The red dashed lines are the median latitudinal location of the corresponding modern potential megabiomes, derived from grid-cells including pollen samples at 0 cal. ka BP. The black solid line represents the median latitude for each timeslice, while the top and lower boundaries of each box represent the upper and lower quartiles of latitude distribution for that timeslice. Megabiome code: TRFO - tropical forest, WTFO - subtropical forest, TEFO - temperate forest, BOFO - boreal forest, SAVA - (warm) savanna and dry woodland, STEP - grassland and dry shrubland, DESE - (warm) desert, TUND - tundra and polar desert.

LGM (represented by the timeslice 21 cal. ka BP): TUND and BOFO dominate the high latitudes and periglacial areas (similar to Prentice et al., 2000), whereas the relatively warm forest megabiomes (e.g., WTFO and TEFO) are distributed at lower latitudes than present-day, in response to cold and dry climates (Nolan et al., 2018). However, the ESM-representative megabiome (simulations hereafter in this Sect.) reveals more non-forest megabiomes (such as TUND and STEP) in periglacial areas of North America (e.g., Alaska and the Rocky Mountains) and northern Asia (e.g., northeastern Siberia), as well as in the Mediterranean regions, as compared to the reconstructions. Although previous pollen-based biomization studies with different biomization schemes have reported ESM-like results (such as Binney et al., 2017 and Cao et al., 2019 in periglacial areas; Elenga et al., 2000 and Prentice et al., 2000 in the Mediterranean regions), assessments of modern megabiome distributions suggest that these studies overestimated the occurrence of non-forest megabiomes in these regions. A recent pollen-based forest cover reconstruction by Davis et al. (2024) indicates more forest than previously suggested by biome reconstructions in these regions during the LGM, which aligns with our results. Furthermore, STEP occurred in central Asia in the reconstructions rather than TUND in the simulations, and TRFO and SAVA appeared in tropical South America and Africa in the reconstructions rather than WTFO in the simulations.

Deglaciation (represented by the timeslices 16 and 13 cal. ka BP): Compared with the LGM, the extratropical megabiomes experienced a remarkable expansion to higher latitudes that coincided with the retreat of the continental ice sheets (Fig. 3). In particular, BOFO, TUND, and TEFO underwent a more extensive expansion compared to the other megabiomes in both our reconstructions and simulations; a result similar to previous biomization studies (such as Binney et al., 2017 and Cao et al., 2019 north of 30°N). However, in contrast to the expansion of forest megabiomes (mostly TEFO and BOFO) in the reconstructions of the Rocky Mountains, northeastern Siberia, and the Mediterranean regions, more non-forest megabiomes (mostly STEP and TUND) occurred in the simulations. TRFO and SAVA expanded in the reconstructions of tropical South America and Africa, whereas the simulations show a shift from WTFO to TRFO since the LGM. In Australia, the Great Dividing Range region was dominated by WTFO in the reconstructions and STEP in the simulations.

Early Holocene (represented by the timeslice 9 cal. ka BP): By this time, the global spatial patterns of megabiome distributions have shifted to closely resemble those of the present-day. That is, forest megabiomes replaced the glacial non-forest megabiomes during the early Holocene and expanded to similar distributional positions as those of today. For example, as the ice sheets receded in the Northern Hemisphere, BOFO continued to move northward and dominated the northern Rockies during the early Holocene, with distributions comparable to today, inferred from both reconstructions and simulations. Due to the extended and homogenized dataset used here, our study also challenges the previous regional-based views that similar distribution patterns of modern megabiomes (Binney et al., 2017) and maximum forest expansion occurred in the mid-Holocene (Ni et al., 2014; Tian et al., 2018). However, mismatches persist between our reconstructions and simulations. For example, Scandinavia was dominated by TEFO and BOFO in the reconstructions but BOFO and TUND in the simulations; Alaska and the Mediterranean regions shifted to BOFO and TEFO, respectively, in the reconstructions, while TUND and STEP remained dominant in the simulations.

Mid-Holocene to Late Holocene (represented by the timeslices 6 and 3 cal. ka BP): The spatial patterns of megabiome distributions during this period are only slightly different from those of the early Holocene. TRFO,

for example, expanded in Mesoamerican reconstructions and simulations. It is also worth noting that the forest megabiomes have not obviously shifted since the Late Holocene, as revealed by both reconstructions and simulations. Given that the simulated vegetation was in a quasi-equilibrium with the climate and unaffected by humans, this implies a relatively stable climate in that period. Therefore, we propose that enhanced anthropogenic disturbances over this time period did not promote forest degradation at a broad spatial scale, and that biomization is robust regarding these disturbance (Prentice et al., 1996; Gotanda et al., 2008).

### 3.3 Comparison of pollen-based and ESM-based simulated megabiome reconstructions

To identify regions and periods with the largest deviations between pollen- and model-derived megabiome distributions, as well as to infer regional contributions to such deviations, we calculated their Earth mover's distances (EMDs; Chevalier et al., 2023b) at each available timeslice and grid-cell (Fig. 4a). Following that, we aggregated the EMD time-series over all grid-cells into 15 regional clusters (Fig. 4b) and synthesized the median EMDs over these regional clusters as representative of the global mean dynamic.

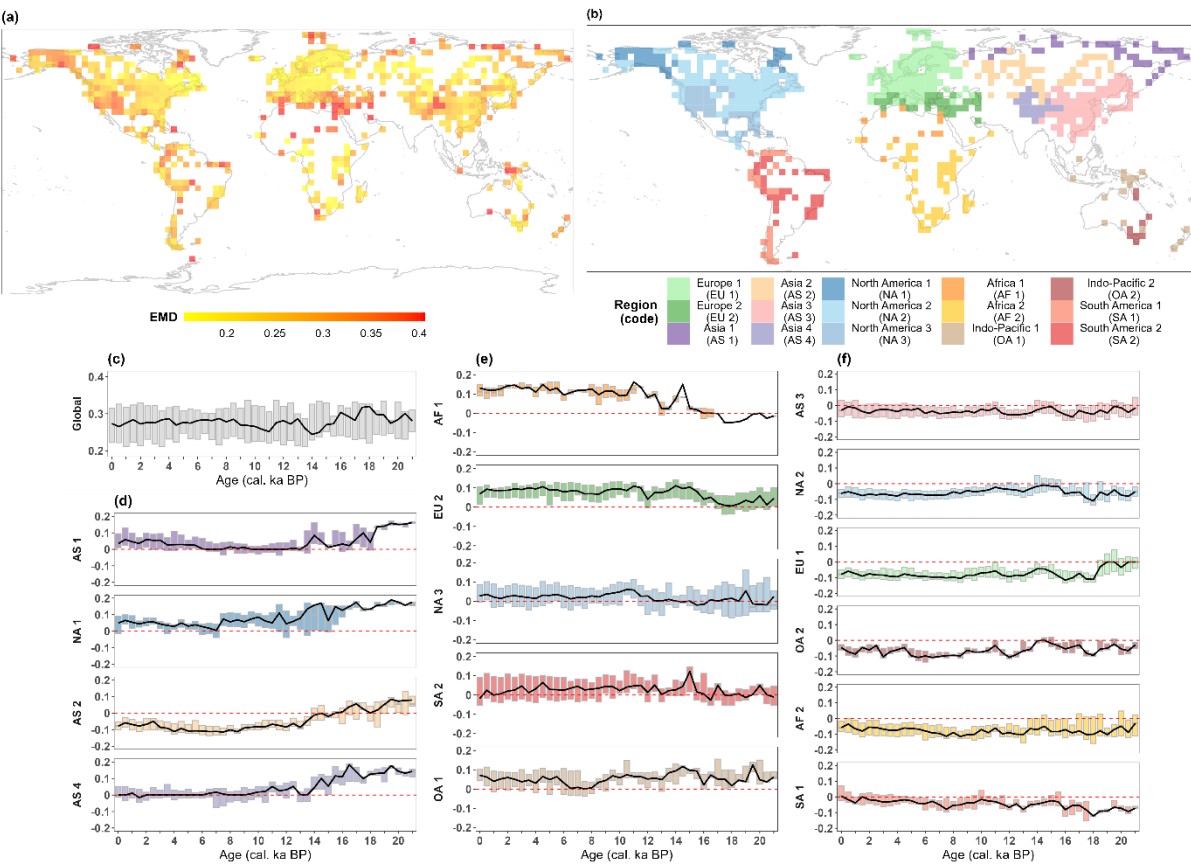

**Figure 4. Spatiotemporal patterns of Earth mover's distance (EMD) between the pollen-based reconstructions and ESM-based simulation ensemble over the last 21,000 years, based on grid-cells of 3.75°x3.75°. (a)** Spatial pattern of the median data-model EMD of available timeslices over the last 21,000 years. Highest EMD values and consequently largest data-model deviations occur especially in the Mediterranean, North Africa, highlands (such as the Rocky Mountains), and circum-Arctic areas. Note that the map legend shows EMD values from the 5[th] to 95[th] percentile, with values above the 95[th] percentile shown in the 95[th] percentile color and values below the 5[th] percentile in the 5[th] percentile color. **(b)** Regional clustering of the data-model EMD time-series for grid-cells using Dynamic Time Warping by continent. **(c)** The global data-model EMD

at each timeslice, synthesized from the median EMDs of the clustered regions at that timeslice. The solid black line represents the median EMD for each timeslice, while the top and lower boundaries of each box represent the upper and lower quartiles of EMD distribution for that timeslice. **(d-f)** The data-model EMD as an anomaly to the global median in clustered regions at each timeslice. That is, regions with **(d)** the highest data-model EMD during the LGM and the early deglaciation, **(e)** the data-model EMD that increases with time during the Holocene, and **(f)** the lower data-model EMD than the global level. Colors and region codes in the boxplots correspond to the colors and region codes of the clusters displayed on the map. The red dashed line is the zero value of EMD.

The largest EMD-assessed deviations between pollen- and model-derived megabiome distributions on a global scale occur during the LGM and early deglaciation (~21–16 cal. ka BP; Fig. 4c). In contrast, the best data-model agreement occurs during the Bølling-Allerød interstadial (represented by the timeslice 14 cal. ka BP) and Early Holocene periods (represented by the timeslice 11 cal. ka BP). Furthermore, the global median EMD has stayed relatively constant at moderate values over the last 9,000 years.

A closer look at the data-model EMD dynamics of the 15 regions (Fig. 4b) identified by the dynamic time warping reveals three sub-clusters. First, regions in which the data-model EMD is particularly high during the LGM and the early deglaciation (Fig. 4d), driving the strong global data-model mismatch during this period. Second, regions in which the data-model EMD rather increases with time (Fig. 4e), contributing to the moderate global EMD values during the Holocene. Third, regions in which the data-model EMD are predominantly lower than the global median EMD (Fig. 4f), i.e., high data-model agreement. However, the reasons for the regional data-model mismatch are very different.

Different estimates of tundra in the circum-Arctic areas and the Tibetan Plateau are the primary sources of the strong global data-model deviations during the LGM and early deglaciation periods (Fig. 4d) at 21 and 16 cal. ka BP (Fig. 3). We observe inconsistent estimates of tundra (TUND) and boreal forest (BOFO) from the pollen-based reconstructions and the ESM-based simulations in northern Siberia (AS1), Alaska (NA1), and the East Siberian Highlands (AS2). To some extent, this mismatch could be attributed to systematic model biases in the simulated climate, as climate models tend to underestimate summer temperature in the periglacial areas compared to proxy-based reconstructions, as previously indicated in studies with different models (Deplazes et al., 2013; Alley, 2000) for that period. The simulations used in this study, especially the MPI-ESM and TRACE-21K simulations, also share this rather common problem in modern times, i.e. a summer cold bias in boreal latitudes (Fig. A3a and Table A1), resulting in an overestimation of tundra in the simulations. However, CLIMBER-X simulations perform better in these regions because they overestimate summer temperatures and produce more boreal forests. Furthermore, the large data-model deviations on the Tibetan Plateau (AS4) result from different estimates of tundra and grasslands (STEP) in the simulations and reconstructions. Given that, the simulated megabiome in the Tibetan plateau that area at timeslice 0 cal. ka BP closely resembles modern potential natural vegetation distributions when compared to the reconstructions (Fig. 1 and Fig. A2), we assume that tundra may have been misrepresented as grassland in the reconstructions.

Different estimates of non-forest megabiomes in relatively semi-arid zones, such as North Africa and the Mediterranean, have contribute to moderate but increasing data-model deviations since the early deglaciation (Fig. 4e). As shown in Fig. 3, with the transition from the glacial to the Holocene, the Mediterranean-Black Sea-Caspian

Corridor (EU2) and the Mediterranean coast of northern Africa have gradually been dominated by temperate forests (TEFO) in the reconstructions, rather than grasslands and dry shrublands (STEP) in the simulations. Since the reconstructions better reproduces the region's modern potential natural vegetation than the simulation (Table 3), we infer that the simulations likely underestimated the cover fraction of woody PFTs in the simulations throughout the Holocene. Given that anthropogenic disturbances (e.g., land use and deforestation) did not promote large-scale forest degradation in this region (cf. Sect. 3.2), this underrepresentation could be attributed to the systematic model biases of hotter summers and drier winters (García-Herrera and Barriopedro, 2018; Fig. A3a–b). In addition, data-model deviations in the Sahara (AF1) are primarily observed during the Holocene, resulting from a mismatch between simulated deserts (DESE) and reconstructed savanna (SAVA). In the simulations, the weakening of the North African monsoon system led to desert expansion in response to seasonal insolation changes, a pattern supported by both proxy-based reconstructions (deMenocal et al., 2000; Shanahan et al., 2015) and climate simulations (Dallmeyer et al., 2021). However, in our reconstructions, the overrepresentation of woody taxa (e.g., *Acacia* and Arecaceae) resulted in the classification of some desert regions as savanna and dry woodlands (SAVA), potentially contributing to the increasing data-model deviations in the Sahara during the Holocene.

**4. Conclusions**

This study presents a global megabiome reconstruction for 43 timeslices at 500-year intervals over the past 21,000 years, based on the most extensive taxonomically and temporally standardized fossil pollen dataset. The dataset's reliability is supported by a high agreement (~80%) with modern potential natural vegetation, and its general consistency with the simulated paleosimulation ensemble further underscores its robustness for exploring past biome dynamics. With its high temporal and spatial coverage, it offers an unprecedented resource, not only for exploring long-term vegetation dynamics and their drivers, but also for diverse research contexts, including paleoclimate, biodiversity, and land-use studies. Furthermore, the dataset supports the evaluation of ESM-based paleo-megabiome simulations and offers insights for identifying potential biases in climate and vegetation models. Its consistent structure and broad applicability allow us to advance our integrative understanding of past, present, and future Earth system dynamics.

**Data availability**

The LegacyPollen 2.0 dataset is open access at PANGAEA (https://doi.org/10.1594/PANGAEA.965907; Li et al., 2025) and provides both count and percentage pollen data. The dataset files in machine-readable data format (.csv) are published in separate data collections into western North America (west of 105°W; Williams et al., 2000), eastern North America, Europe, Asia, South America, Africa, and the Indo-Pacific for easy access and use. We have provided an overview table of record metadata and the taxa harmonization table at PANGAEA, as in the LegacyPollen 1.0 dataset (Herzschuh et al., 2021, 2022).

The simulation MPI-ESM_ICE6G and an equivalent simulation to MPI-ESM_GLAC1D for the biomization tool are available from the Word Data Centre of Climate at https://doi.org/10.26050/WDCC/PMMXMCRTDIP122 (last access: May 16, 2024; Mikolajewicz et al., 2023) and https://doi.org/10.26050/WDCC/PMMXMCHTD (last access: May 16, 2024; Kleinen et al., 2023), respectively. The input data of TRACE-21k-I and TRACE-21k-II for the biomization tool can be downloaded from https://www.earthsystemgrid.org/project/trace.html (last access: May 16, 2024) and https://trace-21k.nelson.wisc.edu/portal.html (last access: May 16, 2024), respectively. The CLIMBER-X simulation is not published, but the input data for the biomization tool can be provided upon request.

The data of modern potential natural vegetation distributions estimated by Ramankutty et al. (2010) can be downloaded from https://daac.ornl.gov/cgi-bin/dsviewer.pl?ds_id=961 (last access: May 16, 2024). The climate dataset of Climatic Research Unit gridded Time Series (CRU TS Version 4.08; Harris et al., 2020) can be downloaded from https://crudata.uea.ac.uk/cru/data/hrg/cru_ts_4.08/cruts.2406270035.v4.08/ (last access: December 22, 2024). The ice-sheet data for ICE-5G (Peltier, 2004) and ICE-6G (Peltier et al., 2015) reconstructions can be downloaded from http://www.atmosp.physics.utoronto.ca/~peltier/data.php (last access: May 16, 2024), and for GLAC-1D (Tarasov et al., 2012) reconstructions can be downloaded from https://pmip4.lsce.ipsl.fr/doku.php/data:ice_glac_1d#download (last access: May 16, 2024).

**Code availability**

We performed all statistical analyses and visualization in this study in the R software environment, and the R scripts have been deposited in the GitHub publication repository (https://github.com/PolarTerrestrialEnvironmentalSystems/Biome; last access: December 24, 2024). The pollen-based biomization algorithm in R and the tool for the biomization of simulated PFT cover fractions are available from Zenodo (https://doi.org/10.5281/zenodo.7523423, last access: May 10, 2023; Cao and Tian, 2021) and MPG.PuRe repository (https://hdl.handle.net/21.11116/0000-0001-B800-F, last access: May 16, 2024; Dallmeyer et al., 2019), respectively. All packages (e.g., '*neotoma2*', '*paleotools*', and '*TSclust*') mentioned throughout are software extensions to R (version 4.4.1; R Core Team, 2023).

**Author contributions**

UH, CL, and AD designed the study. CL and AD performed pollen-based reconstruction and model-based biomization, respectively. CL, JN, and A.A. revised and updated the taxa-PFTs-megabiome assignment schemes in the biomization procedures under the supervision of UH. CL implemented the analysis under the supervision of UH and AD. MW provided the CLIMBER-X simulation. MC and LS contributed to the analytical methods. XC contributed an initial R script for biomization procedures. BH together with MW supported the PANGAEA data publiication of the LegacyPollen 2.0 dataset. CL wrote the first draft of the manuscript under the supervision of UH and AD. All co-authors discussed the results and contributed to the final manuscript.

**Competing interests**

UH, MC, and AD are guest members of the editorial board of Climate of the Past for the special issue "Past vegetation dynamics and their role in past climate changes". The authors have no other competing interests to declare.

**Disclaimer**

Publisher's note: Copernicus Publications remains neutral with regard to jurisdictional claims in published maps and institutional affiliations.

**Acknowledgements**

The majority of the fossil pollen data were obtained from the Neotoma Paleoecology Database (https://www.neotomadb.org/, last access: August 31, 2022) and its constituent databases (e.g., APD, EPD, ALPADABA, IPPD, LAPD, and NAPD). The work of data contributors, data stewards, and the Neotoma community is gratefully acknowledged. We would like to express our gratitude to all the palynologists and geologists who, either directly or indirectly, contributed pollen data and chronologies to the dataset. We thank John W. Williams and Thomas Giesecke from the Neotoma Paleoecology Database for their valuable comments (https://doi.org/10.5194/essd-2023-486-CC3 and https://doi.org/10.5194/essd-2023-486-RC2) on the compilation of the LegacyPollen 2.0 dataset.

We thank Thomas Böhmer for his support with the R script revision. We acknowledge Thomas Kleinen, Uwe Mikolajewicz and Marie Kapsch from the Max Planck Institute for Meteorology, and Feng He from the University of Wisconsin–Madison for providing MPI-ESM and TRACE-21K simulations, respectively. We also thank Cathy Jenks for language editing on a previous version of the paper.

**Financial support**

This research has been supported by the European Research Council (ERC Glacial Legacy 772852 to Ulrike Herzschuh) and the PalMod Initiative (01LP1510C to Ulrike Herzschuh). Anne Dallmeyer, Manuel Chevalier, and Matteo Willeit are supported by the German Federal Ministry of Education and Research (BMBF) as a Research for Sustainability initiative (FONA; https://www.fona.de/en, last access: 10 March 2023) through the PalMod Phase II and Phase III project (grant nos. 01LP1920A, 01LP2306A (AD), and 01LP1926D, 01LP2308B (MC), and 01LP1920B, 01LP1917D, 01LP2305B (MW)). Chenzhi Li holds a scholarship from the Chinese Scholarship Council (grant no. 201908130165).

**Appendix A**

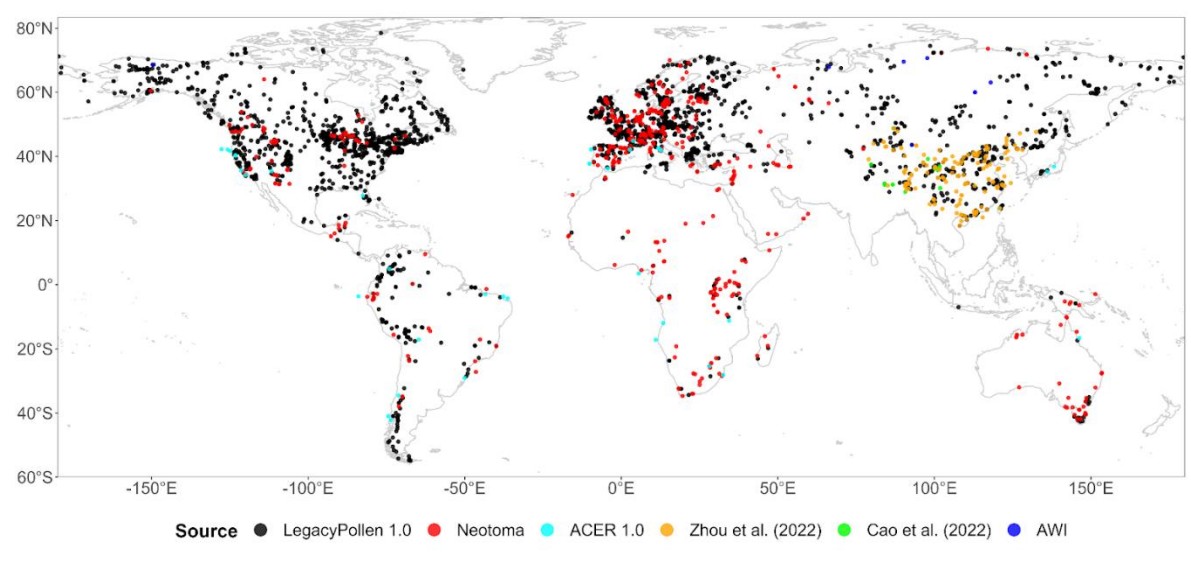

**Figure A1. Spatial distribution and sources of fossil pollen records in the LegacyPollen 2.0 dataset.**

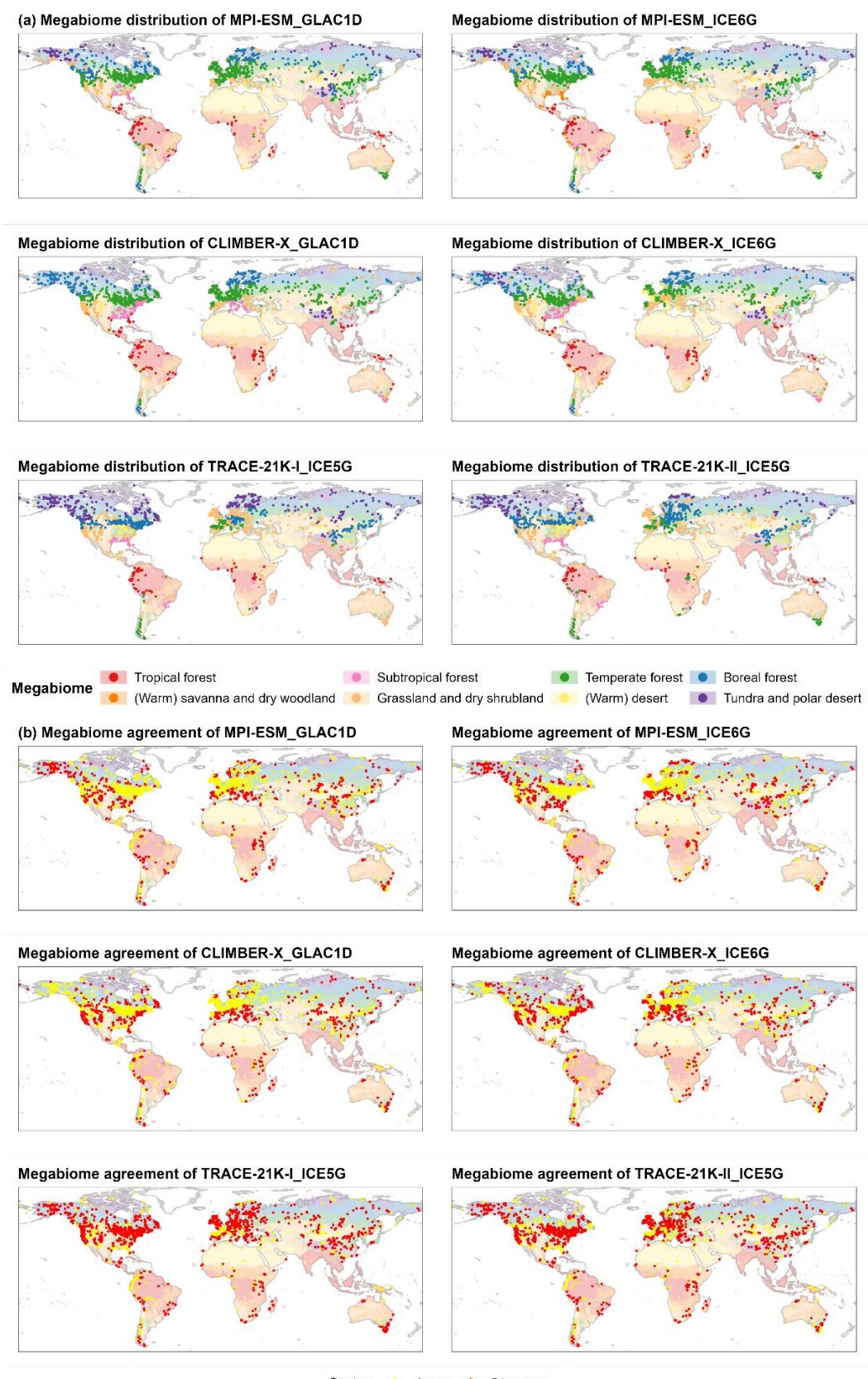


**Figure A2. Spatial patterns of megabiome distributions at 0 cal. ka BP (upper) and their agreement with modern potential natural megabiomes (lower), derived from the ESM-based simulations of MPI-ESM, CLIMBER-X, and TRACE-21K.** The map background depicts the distribution of modern potential natural megabiomes aggregated from modern potential natural vegetation (spatial resolution: 5 arc minutes; Ramankutty et al., 2010; Dallmeyer et al., 2019).

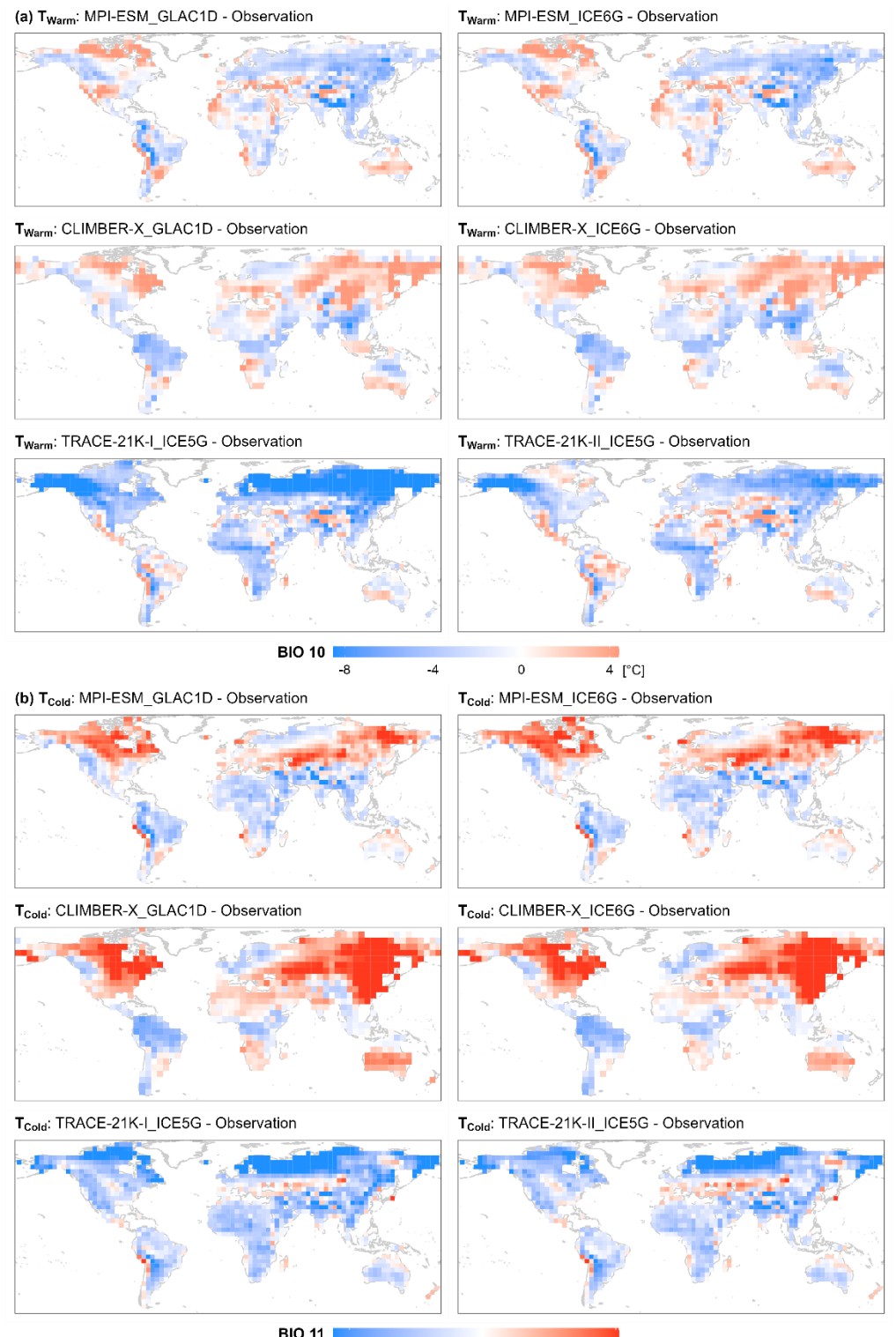

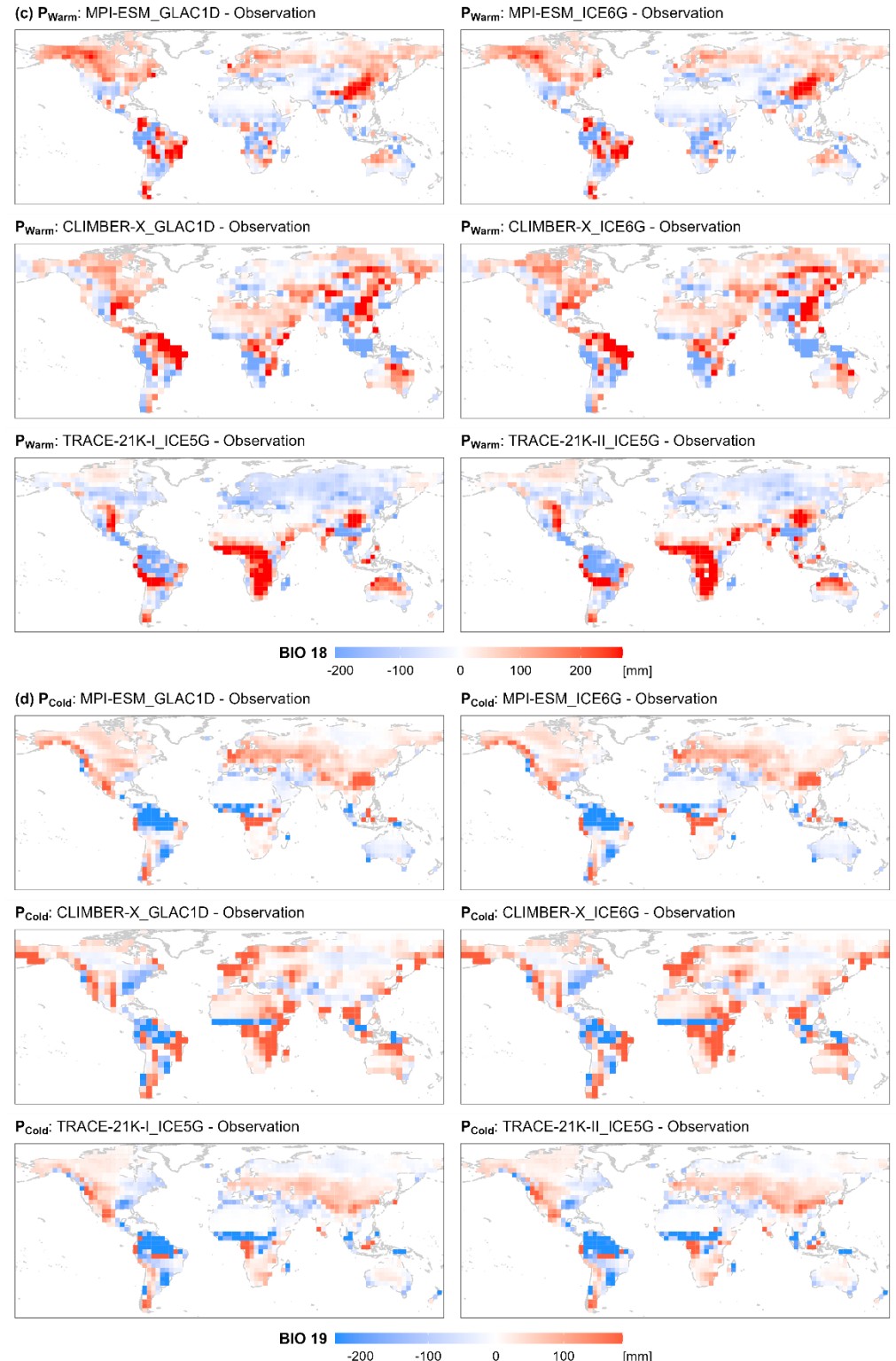

**Figure A3. Differences in bioclimatic variables between ESM-based simulations at 0 cal. ka BP and observations.** The bioclimatic variables include **(a)** mean temperature of warmest quarter ($T_{warm}$), **(b)** mean temperature of coldest quarter ($T_{cold}$), **(c)** precipitation of warmest quarter ($P_{warm}$), and **(d)** precipitation of coldest quarter ($P_{cold}$). The spatial resolutions are 3.75 degrees for the MPI-ESM and TRACE-21K models and 5 degrees for the CLIMBER-X model. Note that the map legend shows bioclimatic variable values from the 5[th] to 95[th] percentile, with values above the 95[th] percentile shown in the 95[th] percentile color and values below the 5[th] percentile in the 5[th] percentile color.


**Table A1. The median difference in bioclimatic variables between ESM-based simulations at 0 cal. ka BP and observations by regions.** The regional clustering is shown in Figure 4b. Bioclimatic variables: $T_{warm}$ - mean temperature of warmest quarter, $T_{cold}$ - mean temperature of coldest quarter, $P_{warm}$ - precipitation of warmest quarter, and $P_{cold}$ - precipitation of coldest quarter. A positive sign in the simulation ensemble difference indicates that the number of simulations that overestimate the bioclimatic variable is greater than the number that underestimate it among the six simulations, while a negative sign indicates the opposite, and positive/negative signs indicate that they are equivalent. Confidence among the six simulations is indicated by one, two, and three asterisks for four, five, and six simulations sharing the same sign, respectively.

| Regions | Bioclimatic variables | MPI-ESM | | CLIMBER-X | | CCSM3 | | Simulation ensemble | |
|---|---|---|---|---|---|---|---|---|---|
| | | MPI-ESM_GLAC1D | MPI-ESM_ICE6G | CLIMBER-X_GLAC1D | CLIMBER-X_ICE6G | TRACE-21K-I_ICE5G | TRACE-21K-II_ICE5G | Difference | Confidence |
| Asia 1 | $T_{warm}$ | -2.4 | -2.3 | 3.1 | 3.6 | -10.0 | -5.5 | - | * |
| | $T_{cold}$ | 1.9 | 3.1 | 4.3 | 4.7 | -10.4 | -9.9 | + | * |
| | $P_{warm}$ | 36.3 | 26.6 | 43.9 | 52.2 | -25.9 | -1.1 | + | * |
| | $P_{cold}$ | 10.5 | 12.1 | 26.6 | 23.7 | -3.0 | 1.8 | + | ** |
| North America 1 | $T_{warm}$ | -0.4 | -0.4 | 0.8 | 0.7 | -8.2 | -5.4 | - | * |
| | $T_{cold}$ | 3.3 | 4.7 | 3.4 | 3.7 | -6.3 | -5.5 | + | * |
| | $P_{warm}$ | 81.9 | 55.4 | 24.7 | 29.7 | -12.7 | 12.4 | + | ** |
| | $P_{cold}$ | 31.6 | 30.7 | 64.8 | 58.8 | 18.7 | 20.7 | + | *** |
| Asia 2 | $T_{warm}$ | -3.8 | -3.6 | 1.7 | 2.0 | -8.2 | -4.1 | - | * |
| | $T_{cold}$ | 2.7 | 3.3 | 6.6 | 6.4 | -5.2 | -3.2 | + | * |
| | $P_{warm}$ | 34.7 | 19.0 | 14.2 | 38.3 | -71.3 | -53.3 | + | * |
| | $P_{cold}$ | 29.3 | 25.9 | -14.2 | -6.7 | 6.7 | 13.6 | + | * |
| Asia 4 | $T_{warm}$ | 0.7 | 1.1 | -0.8 | 0.6 | 2.0 | 2.4 | + | ** |
| | $T_{cold}$ | 0.4 | 0.5 | -0.3 | 0.4 | -2.3 | -2.5 | +/- | |
| | $P_{warm}$ | 0.7 | -25.3 | -33.0 | -33.5 | -4.3 | -0.6 | - | ** |
| | $P_{cold}$ | 51.2 | 49.5 | -0.4 | 1.1 | 70.3 | 79.4 | + | ** |
| Africa 1 | $T_{warm}$ | -0.6 | -0.2 | 0.8 | -0.5 | -1.8 | -1.0 | - | ** |
| | $T_{cold}$ | -0.8 | -1.2 | 1.9 | 0.8 | -1.5 | -1.0 | - | * |
| | $P_{warm}$ | -13.6 | -13.6 | 23.7 | 24.3 | 2.8 | 5.4 | + | * |
| | $P_{cold}$ | -0.5 | -0.5 | 13.2 | 14.0 | -2.3 | -2.4 | - | * |
| Europe 2 | $T_{warm}$ | 0.9 | 0.5 | 0.0 | -0.9 | -2.8 | -1.0 | - | * |
| | $T_{cold}$ | 1.1 | 1.4 | 2.0 | 1.3 | -0.5 | 0.8 | + | ** |
| | $P_{warm}$ | -17.9 | -18.1 | 38.8 | 36.4 | -5.6 | -4.6 | - | * |
| | $P_{cold}$ | -26.2 | -52.3 | -44.0 | -43.6 | -46.4 | -42.9 | - | *** |
| North America 3 | $T_{warm}$ | 0.1 | 0.3 | -0.4 | 0.0 | -1.7 | -1.0 | +/- | |
| | $T_{cold}$ | -0.7 | -0.6 | 1.3 | 0.6 | -1.5 | -0.7 | - | * |
| | $P_{warm}$ | -18.2 | -15.5 | 31.4 | 21.3 | -23.1 | -12.2 | - | * |
| | $P_{cold}$ | 50.3 | 52.9 | 5.2 | 8.1 | 39.6 | 47.4 | + | *** |
| South America 2 | $T_{warm}$ | -1.8 | -1.7 | -3.3 | -3.1 | -0.7 | -0.3 | - | *** |
| | $T_{cold}$ | -2.1 | -1.7 | -2.9 | -3.0 | -1.6 | -1.7 | - | *** |
| | $P_{warm}$ | 125.6 | 96.7 | -45.1 | 57.0 | 7.7 | 14.7 | + | ** |
| | $P_{cold}$ | -13.5 | 2.1 | -0.8 | -0.9 | -75.3 | -89.7 | - | ** |
| Indo-Pacific 1 | $T_{warm}$ | -1.0 | -1.2 | 0.7 | 0.3 | 0.5 | 0.0 | +/- | |
| | $T_{cold}$ | 0.6 | 0.7 | 0.4 | 0.1 | 0.2 | 0.0 | + | ** |
| | $P_{warm}$ | 30.6 | -16.8 | -155.9 | -151.5 | 16.7 | -16.0 | - | * |
| | $P_{cold}$ | -45.2 | -38.8 | -89.8 | -87.2 | -119.1 | -106.4 | - | *** |

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
