# Peer review of "Global biome changes over the last 21,000 years inferred from model-data comparisons"

_EGUsphere, 2024_

## Author Comment (AC1)

**Global biome changes over the last 21,000 years inferred from model-data comparisons**

**Response to comments of Referee #1 Guiot Joel**

**1. General comments**

**Reviewer comment: (1)** *Authors present a new global dataset of 3691 pollen records that are transformed into biomes using the well-established method of Prentice et al (1996). The dataset covers the whole globe with some gaps in Africa, southern Asia and Australia, where the data are not available.*

**Response:** We acknowledge the limited availability of data in certain regions and have clarified this in the text.

**Text:** "While there are geographical gaps in pollen record coverage, particularly in the Southern Hemisphere, the dataset LegacyPollen 2.0 covers the world's main vegetation and climate zones."

**Reviewer comment: (2)** *They work at the level of 8 mega biomes which represent raw pattern of global vegetation. Indeed, it is difficult to work with finer biomes at a global scale (even if Prentice at al 2000 did for two time slices). The biomes reconstructed from 21 ka BP to present at 500 years' time steps are compared with equivalent biomes simulated by three models (and two simulations each).*

**Response:** Thank you for your comment. We have revised the text to clarify the classification of megabiomes and its application in data-model comparison as follows:

**Revised text**: (in red)

"The PFTs were then assigned to megabiomes, representing the raw pattern of global vegetation rather than the finer biomemegabiome categories commonly used in standard biomization studies (Dallmeyer et al., 2019). These megabiomes include), namely tropical forest (TRFO), warm-temperate (subtropical) forest (WTFO), temperate forest (TEFO), boreal forest (BOFO), (warm) savanna and dry woodland (SAVA), grassland and dry shrubland (STEP), (warm) desert (DESE), tundra and polar desert (TUND). These categories wereare also applied to biomize Earth System Model results, which generally use different types and numbers of PFTs to represent global vegetation, enablingallowing for direct data-model comparisons and evaluations (Dallmeyer et al., 2019)."

**Reviewer comment: (3)** *The main problem of the paper can be easily detected in the abstract (which reflect correctly the full paper). The paper is technical and too descriptive.*

**Response:** Thank you for your comment. We have revised the abstract to better emphasize the key findings and broader implications while reducing technical descriptions.

**Revised text:**

"We present a global megabiome reconstruction for 43 timeslices at 500-year intervals throughout the last 21,000 years based on an updated and thus currently most extensive global taxonomically and temporally standardized fossil pollen dataset of 3,455 records. The evaluation with modern potential natural vegetation distributions yields an agreement of ~80%, suggesting a high degree of reliability of the pollen-based megabiome reconstruction. With its high temporal and spatial resolution, this reconstruction is a robust dataset for various applications, such as the evaluation of paleo-simulations from Earth System Models (ESMs).

We compare the reconstruction with an ensemble of six different biomized simulations based on transient vegetation simulations performed by ESMs. The global spatiotemporal patterns of megabiomes estimated by the simulation ensemble and reconstructions are generally consistent. That is, there has been a global shift from open glacial non-forest megabiomes to Holocene forest megabiomes since the Last Glacial Maximum (LGM), in line with the general climate warming trend and continental ice-sheet retreat. The shift to a global megabiome distribution generally similar to today's took place during the early Holocene, while enhanced anthropogenic disturbances since the Late Holocene have not altered broad-scale megabiome patterns. However, certain data-model deviations are evident in specific regions and periods, which could be attributed to systematic climate biases in ESMs or biases in the pollen-based biomization method. For example, at a global scale over the last 21,000 years, the largest deviations between reconstructions and simulation ensembles are observed during the LGM and early deglaciation periods. These discrepancies are probably attributed to systematic summer cold biases in the ESMs that lead to an overestimation of tundra in periglacial regions. Additionally, steppes are overrepresented on the Tibetan Plateau, because steppes share dominant characteristic taxa with tundra and are preferential allocated due to fewer plant functional types (PFTs) compared to tundra. Moderate deviations during the Holocene are primarily driven by differing estimates of non-forest megabiomes in the Mediterranean and North Africa, with increasing discrepancies over time. These deviations may result from the underestimation of woody PFT cover in simulations due to systematic biases, such as overly warm summers with dry winters in the Mediterranean, and the overrepresentation of woody taxa in reconstructions, misclassifying deserts as savanna in North Africa."

**Reviewer comment: (4)** *Biomes are reconstructed, simulated with ESM and compared, some biases of the ESM simulations are pointed out and that is all. There are no general messages on the evolution of the vegetation through the 21 millennia.*

**Response:** We have described the global vegetation evolution over the past 21,000 years in section 3.1 as follows:

**Text: "**We present a global assessment of megabiome dynamics and distributions derived from pollen-based reconstructions and ESM-based simulations over the last 21,000 years, with a temporal resolution of 500 years.

Overall, there has been a global shift from open glacial non-forest megabiomes to Holocene forest megabiomes since the LGM (Fig. 2), in line with the general climate warming trend and continental ice-sheet retreat (Fig. 3):

LGM (represented by the timeslice 21 cal. ka BP): TUND and BOFO dominate the high latitudes and periglacial areas (similar to Prentice et al., 2000 and Davis et al., 2024), whereas the relatively warm forest megabiomes (e.g., WTFO and TEFO) are distributed at lower latitudes than present-day, in response to cold and dry climates. [...]"

**Reviewer comment: (5)** *The ESM biases are not discussed by analysing which climatic variables are responsible of them.*

**Response:** Thank you for your comment. We performed further analyses to assign the ESM biases to certain climatic variables. Due to the lack of pollen-independent climate reconstructions at broad spatial and long temporal scales, we cannot determine when and why models exhibit climate biases. Consequently, we limited our comparison to modern observed climate data in the revised text.

**New text in Method:**

"Modern observational climate data provide a crucial foundation for the assessment of climate simulations. The Climatic Research Unit gridded Time Series (CRU TS hereafter), version 4.08, is a widely used modern observational climate dataset covering all land domains of the world except Antarctica (spatial resolution: ~0.5°x0.5° on a Gaussian grid; Harris et al., 2020). The CRU TS dataset is interpolated from extensive networks of weather station observations and provides monthly temperature and precipitation data from 1901-2023 C.E. However, the early record of this dataset may have high uncertainty due to the sparse observation networks (Duan et al., 2024), and the late record is strongly influenced by anthropogenic $CO_2$ increases (Cheng et al., 2022). We, therefore, selected monthly climatological means from 1931-1970 to generate more biologically meaningful bioclimatic variables for evaluating climate simulations at 0 cal. ka BP (O'Donnell and Ignizio, 2012; Supplementary Data 7). These bioclimatic variables represent extreme or limiting environmental factors, namely, mean temperature of warmest quarter ($T_{warm}$), mean temperature of coldest quarter ($T_{cold}$), precipitation of warmest quarter ($P_{warm}$), and precipitation of coldest quarter ($P_{cold}$). Temperature is given in degrees Celsius (°C), precipitation in millimeters (mm), and a quarter is a period of three consecutive months (1/4 of the year)."

**Revised text in Results and Discussion:**

"The agreement between modern potential megabiomes and simulated megabiomes at timeslice 0 cal. ka BP is higher for the ESM-representative megabiome (cf. Sect. 2.3) than for individual ESM-based simulation (64.1% vs. 20.0–60.2%; Table 3). As a result, the ESM-representative megabiome depicts more reliable patterns of megabiome dynamics and distribution than individual simulations, with higher agreement especially in Alaska, the Iberian Peninsula, the Alps, the Atlantic Coastal Plain of North America, and the southeastern United States (Fig. 1 and Fig. A2). However, there are still certain regions with low agreement, probably due to climatic biases. These include nearly all highlands (such as the central-southern Rockies, the central Andes, and the Tibetan

Plateau) for which an overestimation of the temperature can be expected in the models due to a much lower orography than in reality caused by the smoothing in the coarse spatial resolution (3.75°x3.75° and 5°x5°) of the model grids (Fig. A3a–b). All models simulate non-forest megabiomes instead of forest in the Mediterranean region, which can be attributed to the models simulating a climate that is too seasonally dry, with, for example, too-warm summers and too-dry winters (Fig. A3a, d). The TRACE-21K simulation as well as the MPI-ESM simulations fail to reproduce the boreal forest (BOFO) in Alaska, which is then also reflected in the ESM-representative megabiomes. This failure is likely due to the simulated climate being too cold in this region, preventing the establishment of boreal forests under modeled conditions (Fig. A3a, d). Similar to the reconstructions, the transition zones between temperate forest (TEFO) and non-forest megabiomes, such as the East Asian summer monsoon margin, are regions with lower simulated megabiome agreement to the modern potential megabiome distribution. In North Africa, the models also tend to underestimate the northern extension of the grassland and dry shrubland (STEP) and incorrectly assign (warm) savanna and dry woodland (SAVA) records to tropical forest (TRFO). This is related to the biomization procedure for the model results that only relies on simulated vegetation cover fractions and simulated climate, whereas savannas are additionally determined by other ecological processes such as fire intensity and frequency (Dallmeyer et al., 2019) or grazing (van Langevelde et al., 2019)."

**New figure (Appendix A3):**

[Figure]

**(a)** T_Warm: MPI-ESM_GLAC1D - Observation
T_Warm: MPI-ESM_ICE6G - Observation
T_Warm: CLIMBER-X_GLAC1D - Observation
T_Warm: CLIMBER-X_ICE6G - Observation
T_Warm: TRACE-21K-I_ICE5G - Observation
T_Warm: TRACE-21K-II_ICE5G - Observation

**BIO 10**
-8    -4    0    4    [°C]

**(b)** T_Cold: MPI-ESM_GLAC1D - Observation
T_Cold: MPI-ESM_ICE6G - Observation
T_Cold: CLIMBER-X_GLAC1D - Observation
T_Cold: CLIMBER-X_ICE6G - Observation
T_Cold: TRACE-21K-I_ICE5G - Observation
T_Cold: TRACE-21K-II_ICE5G - Observation

**BIO 11**
-8    -4    0    4    8    [°C]

[Figure]

**(c) P_Warm**: MPI-ESM_GLAC1D - Observation

P_Warm: MPI-ESM_ICE6G - Observation

P_Warm: CLIMBER-X_GLAC1D - Observation

P_Warm: CLIMBER-X_ICE6G - Observation

P_Warm: TRACE-21K-I_ICE5G - Observation

P_Warm: TRACE-21K-II_ICE5G - Observation

**BIO 18**
-200  -100  0  100  200  [mm]

**(d) P_Cold**: MPI-ESM_GLAC1D - Observation

P_Cold: MPI-ESM_ICE6G - Observation

P_Cold: CLIMBER-X_GLAC1D - Observation

P_Cold: CLIMBER-X_ICE6G - Observation

P_Cold: TRACE-21K-I_ICE5G - Observation

P_Cold: TRACE-21K-II_ICE5G - Observation

**BIO 19**
-200  -100  0  100  [mm]

**Figure A3. Differences in bioclimatic variables between ESM-based simulations at 0 cal. ka BP and observations.** The bioclimatic variables include **(a)** mean temperature of warmest quarter ($T_{warm}$), **(b)** mean temperature of coldest quarter ($T_{cold}$), **(c)** precipitation of warmest quarter ($P_{warm}$), and **(d)** precipitation of coldest quarter ($P_{cold}$). Notable biases include overestimated temperatures ($T_{warm}$ and $T_{cold}$) in highlands (e.g., Rockies, Andes, Tibetan Plateau), excessively dry Mediterranean summers ($P_{warm}$), and colder-than-observed conditions in Alaska ($T_{warm}$ and $T_{cold}$).

**Reviewer comment: (6)** *Why mediterranean biomes of the Holocene are simulated as steppes whereas data indicate TEDE?*

**Response:** All models consistently simulate steppe (STEP) rather than temperate forest (TEDE) in the Mediterranean region during the Holocene, likely due to systematic biases in the simulated climate, as described in the revised text.

**Revised text:**

**"**The Mediterranean region has warm to hot dry summers and mild wet winters. Modeling studies report systematic model biases of too-warm summers and too-dry winters in this region (García-Herrera and Barriopedro, 2018). A comparison with modern data shows similar climate biases in the simulations which may indicate similar systematic biases in the past. This would explain the underrepresentation of the cover fraction of woody PFTs in the simulations (Fig. A3a, d).**"**

**Reviewer comment: (7)** *There are also biases in the reconstructed biomes. As an example, the glacial mediterranean biomes are sometimes reconstructed as TEDO while previous reconstructions (Elenga et al, 2000; Prentice et al, 2000) reconstructed STEP.*

**Response:** Thank you for your comment. We have added more comparisons in the revised text to clarify the differences between our reconstructions and previous studies.

**Revised text:** (in red)

**"**LGM (represented by the timeslice 21 cal. ka BP): TUND and BOFO dominate the high latitudes and periglacial areas (similar to Prentice et al., 2000), whereas the relatively warm forest megabiomes (e.g., WTFO and TEFO) are distributed at lower latitudes than present-day, in response to cold and dry climates (Nolan et al., 2018). However, the ESM-representative megabiome (simulations hereafter in this Sect.) reveals more non-forest megabiomes (such as TUND and STEP) in periglacial areas of North America (e.g., Alaska and the Rocky Mountains) and northern Asia (e.g., northeastern Siberia), as well as in the Mediterranean regions, as compared to the reconstructions. Although previous pollen-based biomization studies with different biomization schemes have reported ESM-like results (such as Binney et al., 2017 and Cao et al., 2019 in periglacial areas; Elenga et al., 2000 and Prentice et al., 2000 in the Mediterranean regions), assessments of modern megabiome distributions

suggest that these studies overestimated the occurrence of non-forest megabiomes in these regions. A recent pollen-based forest cover reconstruction by Davis et al. (2024) indicates more forest than previously suggested by biome reconstructions in these regions during the LGM, which aligns with our results. Furthermore, STEP occurred in central Asia in the reconstructions rather than TUND in the simulations, and TRFO and SAVA appeared in tropical South America and Africa in the reconstructions rather than WTFO in the simulations."

**Reviewer comment: (8)** *I have the feeling that for the regions I know better, this paper does a poorer reconstruction than previous attempts. If not, this should be argued.*

**Response:** Thank you for your comment. We have added regional comparisons in the revised text to further clarify the reliability of our reconstruction.

**Revised text:**

"We consider global-scale, pollen-based megabiome reconstructions reliable because record-by-record comparisons of reconstructed megabiomes at timeslice 0 cal. ka BP from 2,232 available records with modern potential megabiomes indicate an 80.2% agreement (Table 3). This consistency exceeds that reported in previous large-scale biomization studies validated against modern biome distributions, such as the 53% agreement in Arctic high-latitudes (>55°N) by Bigelow et al. (2003). We assume that the high agreement not only originates from the high quality of the pollen data set used with respect to taxonomical and temporal harmonization, but also relates to the fact that the biomization method employs updated and harmonized schemes assigning pollen taxa to plant functional types to megabiomes. Furthermore, our reconstruction was performed at the megabiome level, a coarser classification than typical biomes, which somewhat reduces the mismatch between geographically adjacent biomes. For example, the biomes of temperate deciduous forest and cool mixed forest are intermingled in Binney et al. (2017), whereas this mismatch does not exist at the megabiome level of temperate forest. Although some regional-scale biomization studies achieve higher agreement with modern biome distributions, such as 97.5% in the Congo Basin by Lebamba et al. (2009), these often rely on more localized datasets with tailored taxa-PFT-biome schemes. As a result, we argue that the data quality as well as the higher spatial and temporal coverage compared to previous biomization studies (Bigelow et al., 2003; Marinova et al., 2018) make our pollen-based megabiome reconstruction a robust dataset for various applications, such as global-scale evaluation of paleo-simulations from Earth System Models (ESMs)."

**Reviewer comment: (9)** *In conclusion, the paper has potentialities, but it needs more work to be a good contribution to the discipline.*

**Response:** We sincerely appreciate the reviewer's thoughtful feedback and recognition of the potential of our work. We have carefully addressed the issues raised in the review, please refer to the previous responses.

**2. Specific comments**

**Reviewer comment: (1)** *Section 2.3: how are interpolated the biomes in sites from the grid of the ESM (the dots of Fig 1B)?*

**Response:** Thank you for your comment. We have revised the text in Section 2.3:

**Revised text:** (in red)

"We assigned the simulated megabiome data taken from the grid-cells where the records are located to each record, and we only considered records and timeslices for which reconstructions are available."

**Reviewer comment: (2)** *230-233: I disagree that there are no systematic mismatch: there are in the Med region, in the subarctic one ... Note also that TUND and STEP have low agreement (50%)*

**Response:** Thank you for your comment. The sentence has been removed, and the lower agreement of TUND and STEP has been clarified in the revised text.

**Revised text:**

"The low taxonomic resolution could also cause mismatches between neighboring forest megabiomes, as well as between tundra (TUND) and grassland (STEP). Woody taxa have been harmonized to the genus level rather than the species level, while herbaceous taxa are generally harmonized to the family level, except for common taxa like *Artemisia*, *Thalictrum*, and *Rumex*. This reduces the ecological information available for PFT assignment (Chen et al., 2010).

[...]

Similarly, TUND may have been misrepresented as STEP on the Tibetan Plateau. This misrepresentation can be attributed to they share dominant characteristic species of Poaceae and Cyperaceae, whereas STEP is defined by fewer PFTs and therefore preferentially allocated. However, the woody PFTs are generally not defined in STEP, leading to a potential misallocation to TUND rather than STEP in cases of woody pollen grain occurrences (from long-distance transportation or local existence) in open landscape samples (Marinova et al., 2018; Chen et al., 2010), such as mismatches in southern Europe."

**Reviewer comment: (3)** *233-235: I think that the good agreement comes to the fact that the comparison is restricted to 8 megabiomes. In the previous papers, finer biomes are considered and often the mismatch is between climatically neighbor biomes (this is less possible with megabiomes).*

**Response:** We agree that the good agreement is partly due to being limited to 8 megabiomes. We have revised the text accordingly.

**Revised text:**

"We consider global-scale, pollen-based megabiome reconstructions reliable because record-by-record comparisons of reconstructed megabiomes at timeslice 0 cal. ka BP from 2,232 available records with modern potential megabiomes indicate an 80.2% agreement (Table 3). This consistency exceeds that reported in previous large-scale biomization studies validated against modern biome distributions, such as the 53% agreement in Arctic high-latitudes (>55°N) by Bigelow et al. (2003). We assume that the high agreement not only originates from the high quality of the pollen data set used with respect to taxonomical and temporal harmonization, but also relates to the fact that the biomization method employs updated and harmonized schemes assigning pollen taxa to plant functional types to megabiomes. Furthermore, our reconstruction was performed at the megabiome level, a coarser classification than typical biomes, which somewhat reduces the mismatch between geographically adjacent biomes. For example, the biomes of temperate deciduous forest and cool mixed forest are intermingled in Binney et al. (2017), whereas this mismatch does not exist at the megabiome level of temperate forest. Although some regional-scale biomization studies achieve higher agreement with modern biome distributions, such as 97.5% in the Congo Basin by Lebamba et al. (2009), these often rely on more localized datasets with tailored taxa-PFT-biome schemes."

**Reviewer comment: (4)** *259-263: I do not understand this sentence.*

**Response:** We apologize for the lack of clarity in the original text. The sentence has been revised for better understanding as follows:

**Revised text:**

"Similarly, TUND may have been misrepresented as STEP on the Tibetan Plateau. This misrepresentation can be attributed to they share dominant characteristic species of Poaceae and Cyperaceae, whereas STEP is defined by fewer PFTs and therefore preferentially allocated. However, the woody PFTs are generally not defined in STEP, leading to a potential misallocation to TUND rather than STEP in cases of woody pollen grain occurrences (from long-distance transportation or local existence) in open landscape samples (Marinova et al., 2018; Chen et al., 2010), such as mismatches in southern Europe."

**Reviewer comment: (5)** *Section 3.1.2: It is necessary to try to explain the discrepancies between simulated biomes and potential vegetation by over or under-simulation of some climate variables.*

**Response:** We have incorporated an additional evaluation using modern climate data as follows:

**Revised text:**

"The agreement between modern potential megabiomes and simulated megabiomes at timeslice 0 cal. ka BP is higher for the ESM-representative megabiome (cf. Sect. 2.3) than for individual ESM-based simulation (64.1% vs. 20.0–60.2%; Table 3). As a result, the ESM-representative megabiome depicts more reliable patterns of megabiome dynamics and distribution than individual simulations, with higher agreement especially in Alaska, the Iberian Peninsula, the Alps, the Atlantic Coastal Plain of North America, and the southeastern United States (Fig. 1 and Fig. A2). However, there are still certain regions with low agreement, probably due to climatic biases. These include nearly all highlands (such as the central-southern Rockies, the central Andes, and the Tibetan Plateau) for which an overestimation of the temperature can be expected in the models due to a much lower orography than in reality caused by the smoothing in the coarse spatial resolution (3.75°x3.75° and 5°x5°) of the model grids (Fig. A3a–b). All models simulate non-forest megabiomes instead of forest in the Mediterranean region, which can be attributed to the models simulating a climate that is too seasonally dry, with, for example, too-warm summers and too-dry winters (Fig. A3a, d). The TRACE-21K simulation as well as the MPI-ESM simulations fail to reproduce the boreal forest (BOFO) in Alaska, which is then also reflected in the ESM-representative megabiomes. This failure is likely due to the simulated climate being too cold in this region, preventing the establishment of boreal forests under modeled conditions (Fig. A3a, d). Similar to the reconstructions, the transition zones between temperate forest (TEFO) and non-forest megabiomes, such as the East Asian summer monsoon margin, are regions with lower simulated megabiome agreement to the modern potential megabiome distribution. In North Africa, the models also tend to underestimate the northern extension of the grassland and dry shrubland (STEP) and incorrectly assign (warm) savanna and dry woodland (SAVA) records to tropical forest (TRFO). This is related to the biomization procedure for the model results that only relies on simulated vegetation cover fractions and simulated climate, whereas savannas are additionally determined by other ecological processes such as fire intensity and frequency (Dallmeyer et al., 2019) or grazing (van Langevelde et al., 2019)."

**Reviewer comment: (6)** *Figure 3: how is obtained the ice-sheet extension data from pollen?*

**Response:** Thank you for your comment. To clarify, the ice-sheet extension data in Figure 3 are not derived from pollen data. Instead, they are based on an ice-sheet ensemble set designed for fair comparisons among simulations.

**Text:** "We also created an ice-sheet ensemble set with a spatial resolution of 3.75°, synthesized from the maximum extent of ICE-5G, ICE-6G, and GLAC-1D reconstructions, for fair comparisons among simulations."

**Reviewer comment: (7)** *320-331: Elenga et al 2000 and Prentice et al 2000 reconstructed steppes at 21 ka BP, which seems more realistic than tundra. Prentice et al 2000 is not cited despite the fact that they produced a full global reconstruction for the 21 ka BP period (and also mid-Holocene). It is a major paper that the authors cannot ignore. 325-332: the problem of assigning TUND to STEP and vice versa should also be discussed; previous papers reconstruct mainly STEP to the Med region and not TUND as here. There are also biases in the reconstructed biomes.*

**Response:** Thanks for your recommendation and we have included this reference. Our reconstruction may have misrepresented STEP as TUND, which has been discussed in the revised text:

**Revised text:**

**"**The low taxonomic resolution could also cause mismatches between neighboring forest megabiomes, as well as between tundra (TUND) and grassland (STEP). Woody taxa have been harmonized to the genus level rather than the species level, while herbaceous taxa are generally harmonized to the family level, except for common taxa like *Artemisia*, *Thalictrum*, and *Rumex*. This reduces the ecological information available for PFT assignment (Chen et al., 2010).

[...]

Similarly, TUND may have been misrepresented as STEP on the Tibetan Plateau. This misrepresentation can be attributed to they share dominant characteristic species of Poaceae and Cyperaceae, whereas STEP is defined by fewer PFTs and therefore preferentially allocated. However, the woody PFTs are generally not defined in STEP, leading to a potential misallocation to TUND rather than STEP in cases of woody pollen grain occurrences (from long-distance transportation or local existence) in open landscape samples (Marinova et al., 2018; Chen et al., 2010), such as mismatches in southern Europe.**"**

**Reviewer comment: (8)** *333-334: TUND seems more extended in Europe at 16-13ka than at 21ka, while the warming has already started, why?*

**Response:** Thank you for your comment. We have clarified this in the revised text:

**Revised text:**

**"**Deglaciation (represented by the timeslices 16 and 13 cal. ka BP): Compared with the LGM, the extratropical megabiomes experienced a remarkable expansion to higher latitudes that coincided with the retreat of the continental ice sheets (Fig. 3). In particular, BOFO, TUND, and TEFO underwent a more extensive expansion compared to the other megabiomes in both our reconstructions and simulations; a result similar to previous biomization studies (such as Binney et al., 2017 and Cao et al., 2019 in north of 30°N). **"**

**Reviewer comment: (9)** *342-353: Biomization starts to be realistic in the Holocene, much more than for the cold periods. But It is strange that there is no Mediterranean vegetation (WTFO) in the Med area during all the Holocene, as well for simulation as reconstruction. For simulation WTFO is replaced by STEP and for reconstruction, it is by TEFO.*

**Response: (a)** All models consistently simulate steppe (STEP) rather than subtropical forest (WTFO) in the Mediterranean region during all the Holocene, likely due to systematic biases in the simulated climate, as described in the revised text.

**Revised text: "**The Mediterranean region has warm to hot dry summers and mild wet winters. Modeling studies report systematic model biases of too-warm summers and too-dry winters in this region (García-Herrera and Barriopedro, 2018). A comparison with modern data shows similar climate biases in the simulations which may indicate similar systematic biases in the past. This would explain the underrepresentation of the cover fraction of woody PFTs in the simulations (Fig. A3a, d).**"**

**(b)** In our reconstruction, we may have misrepresented WTFO as TEFO, which has been clarified in the revised text.

**Revised text:**

**"**The low taxonomic resolution could also cause mismatches between neighboring forest megabiomes, as well as between tundra (TUND) and grassland (STEP). Woody taxa have been harmonized to the genus level rather than the species level, while herbaceous taxa are generally harmonized to the family level, except for common taxa like *Artemisia*, *Thalictrum*, and *Rumex*. This reduces the ecological information available for PFT assignment (Chen et al., 2010). For instance, different species within *Pinus*, *Alnus*, *Fagus*, and *Betula* (Tian et al., 2018) have different bioclimatic controls, phenology, and life forms, but identification at the genus level results in them being shared by key PFTs in different forest megabiomes (e.g., WTFO vs. TEFO, TEFO vs. BOFO) when assigning taxa to PFTs. One of the typical areas in which this problem occurs is southern Scandinavia. Pollen grains from *Betula pendula* in temperate forests and *Betula pubescens* in boreal forests (Beck et al., 2016) in this region can only be identified to genus level, resulting in these two key species not being able to serve as indicators to distinguish between temperate and boreal forests.**"**

**(c)** However, the WFTO is not the dominant potential natural megabiome in the modern Mediterranean region. As shown in the figure below, which depicts the distribution of modern potential natural megabiomes aggregated from modern potential natural vegetation (Ramankutty and Foley, 1999; Ramankutty et al., 2010). It represents the world's vegetation cover that had most likely existed for 1986–1995 C.E. in equilibrium with present-day climate and natural disturbance in the absence of human activities.

[Figure]

**Figure.** Spatial patterns of modern potential natural megabiome distributions.

**Reviewer comment: (10)** *356-361: I read on the maps the opposite to what is claimed: I see a forest degradation in the simulations (maps at right) not in the reconstructions (maps at left). In the whole Holocene, reconstructions show a constant TEFO, while simulation shows steppes in the second part of the Holocene, tending to show that there is a bias towards aridity in the models. Pollen reconstructions may sometimes be influenced by human deforestation, but it has been shown in previous papers that biomization is robust as regards as this disturbance.*

**Response:** We have revised this paragraph based on your suggestion.
**Revised text:**

"Mid-Holocene to Late Holocene (represented by the timeslices 6 and 3 cal. ka BP): The spatial patterns of megabiome distributions during this period are only slightly different from those of the early Holocene. TRFO, for example, expanded in Mesoamerican reconstructions and simulations. It is also worth noting that the forest megabiomes have not obviously shifted since the Late Holocene, as revealed by both reconstructions and simulations. Given that the simulated vegetation was in a quasi-equilibrium with the climate and unaffected by humans, this implies a relatively stable climate in that period. Therefore, we propose that enhanced anthropogenic disturbances over this time period did not promote forest degradation at board spatial scale, and that biomization is robust as regards as this disturbance (Prentice et al., 1996; Gotanda et al., 2008)."

**Reviewer comment: (11)** *Figure 4: It does not exist the possibility to computed significance levels for EDM?*

**Response:** Thank you for the comment. While it is technically feasible to compute significance levels for the EMD, we have chosen not to perform such calculations as they are not directly relevant to the goals of our study.

The Earth Mover's Distance (EMD) is designed to quantify the degree of mismatch between reconstructions and simulations by integrating uncertainties and weighted ecological and climatic distances. Its primary purpose is to provide a nuanced, distribution-based metric for evaluating spatiotemporal patterns of biome differences rather than testing for random variation or statistical significance.

Significance levels typically require assumptions about the underlying data distribution or the application of resampling techniques, such as bootstrapping. However, the weighting scheme we employed in our EMD calculations is highly context-specific, reflecting ecological and climatic gradients rather than stochastic variation. This specificity makes conventional significance testing less informative in our context.

Moreover, the core objective of our analysis is to assess the relative agreement between reconstructions and simulations across regions and timeslices, focusing on pattern visualization and identifying areas with stronger or weaker model-data agreement. Introducing significance levels would not substantially contribute to the interpretability or scientific value of our results, as the threshold would be inconsistent across different spatial and temporal contexts due to variations in data density and spatial autocorrelation.

To ensure methodological clarity, we emphasize the robustness of our approach by employing carefully defined affinity scores and weighting schemes that directly address vegetation-climate dynamics. These components have been selected to ensure that the EMD reflects biologically and climatically meaningful differences, aligning with the central aims of our study.

**Reviewer comment: (12)** *Caption of Fig.4: The sentence "The largest datamodel deviations occur during the LGM and early deglaciation periods" should not be put in the caption.*

**Response:** We've removed it as you suggested.

**Reviewer comment: (13)** *385-389: "the best data-model agreement occurs during the Bølling-Allerød interstadial (represented by the timeslice 14 cal. ka BP)": This appears to be true with the global EMD but in the regional EMD, it does not appear that 14ka EDM was minimum in any regions.*

**Response:** The global data-model EMD at each timeslice is derived from the median EMDs of clustered regions at that timeslice. While the global EMD indicates the best data-model agreement during the Bølling-Allerød interstadial (14 cal. ka BP), this does not necessarily mean that the EMD for this timeslice was the minimum in any specific region. Instead, the global pattern represents the aggregated dynamics across all regions, where regional variations may exhibit significant heterogeneity. This distinction reflects the synthesis approach used in our analysis and highlights the importance of considering both global and regional perspectives.

**Reviewer comment: (14)** *Section 3.3: it should be useful to summarize by a table the biases found in simulations and giving an interpretation of which climate variable is responsible of the biases*

**Response:** We do not have pollen-independent climate reconstructions. So we cannot really say when and why the models have a climate that is too different, causing differences in vegetation compared to the reconstructions. There are some other reconstructions available, but it is not always clear which proxy records which climate variable, and deciding this is beyond our expertise, and a fair comparison of the past model climate with the reconstructions would be a huge effort and beyond the scope of this paper. Therefore, we can only compare the modern observed climate with the models and add a new table to the appendix.

**New table:**

**Table A1. The median difference in bioclimatic variables between ESM-based simulations at 0 cal. ka BP and observations by regions.** The regional clustering is shown in Figure 4b. Bioclimatic variables: $T_{warm}$ - mean temperature of warmest quarter, $T_{cold}$ - mean temperature of coldest quarter, $P_{warm}$ - precipitation of warmest quarter, and $P_{cold}$ - precipitation of coldest quarter. A positive sign in the simulation ensemble difference indicates that the number of simulations that overestimate the bioclimatic variable is greater than the number that underestimate it among the six simulations, while a negative sign indicates the opposite, and positive/negative signs indicate that they are equivalent. Confidence among the six simulations is indicated by one, two, and three asterisks for four, five, and six simulations sharing the same sign, respectively.

| Regions | Bioclimatic variables | MPI-ESM | | CLIMBER-X | | CCSM3 | | Simulation ensemble | |
|---|---|---|---|---|---|---|---|---|---|
| | | MPI-ESM_GLAC1D | MPI-ESM_ICE6G | CLIMBER-X_GLAC1D | CLIMBER-X_ICE6G | TRACE-21K-I_ICE5G | TRACE-21K-II_ICE5G | Difference | Confidence |
| Asia 1 | $T_{warm}$ | -2.4 | -2.3 | 3.1 | 3.6 | -10.0 | -5.5 | - | * |
| | $T_{cold}$ | 1.9 | 3.1 | 4.3 | 4.7 | -10.4 | -9.9 | + | * |
| | $P_{warm}$ | 36.3 | 26.6 | 43.9 | 52.2 | -25.9 | -1.1 | + | * |
| | $P_{cold}$ | 10.5 | 12.1 | 26.6 | 23.7 | -3.0 | 1.8 | + | ** |
| North America 1 | $T_{warm}$ | -0.4 | -0.4 | 0.8 | 0.7 | -8.2 | -5.4 | - | * |
| | $T_{cold}$ | 3.3 | 4.7 | 3.4 | 3.7 | -6.3 | -5.5 | + | * |
| | $P_{warm}$ | 81.9 | 55.4 | 24.7 | 29.7 | -12.7 | 12.4 | + | ** |
| | $P_{cold}$ | 31.6 | 30.7 | 64.8 | 58.8 | 18.7 | 20.7 | + | *** |
| Asia 2 | $T_{warm}$ | -3.8 | -3.6 | 1.7 | 2.0 | -8.2 | -4.1 | - | * |
| | $T_{cold}$ | 2.7 | 3.3 | 6.6 | 6.4 | -5.2 | -3.2 | + | * |
| | $P_{warm}$ | 34.7 | 19.0 | 14.2 | 38.3 | -71.3 | -53.3 | + | * |
| | $P_{cold}$ | 29.3 | 25.9 | -14.2 | -6.7 | 6.7 | 13.6 | + | * |
| Asia 4 | $T_{warm}$ | 0.7 | 1.1 | -0.8 | 0.6 | 2.0 | 2.4 | + | ** |
| | $T_{cold}$ | 0.4 | 0.5 | -0.3 | 0.4 | -2.3 | -2.5 | +/- | |
| | $P_{warm}$ | 0.7 | -25.3 | -33.0 | -33.5 | -4.3 | -0.6 | - | ** |
| | $P_{cold}$ | 51.2 | 49.5 | -0.4 | 1.1 | 70.3 | 79.4 | + | ** |
| Africa 1 | $T_{warm}$ | -0.6 | -0.2 | 0.8 | -0.5 | -1.8 | -1.0 | - | ** |
| | $T_{cold}$ | -0.8 | -1.2 | 1.9 | 0.8 | -1.5 | -1.0 | - | * |
| | $P_{warm}$ | -13.6 | -13.6 | 23.7 | 24.3 | 2.8 | 5.4 | + | * |
| | $P_{cold}$ | -0.5 | -0.5 | 13.2 | 14.0 | -2.3 | -2.4 | - | * |
| Europe 2 | $T_{warm}$ | 0.9 | 0.5 | 0.0 | -0.9 | -2.8 | -1.0 | - | * |
| | $T_{cold}$ | 1.1 | 1.4 | 2.0 | 1.3 | -0.5 | 0.8 | + | ** |
| | $P_{warm}$ | -17.9 | -18.1 | 38.8 | 36.4 | -5.6 | -4.6 | - | * |
| | $P_{cold}$ | -26.2 | -52.3 | -44.0 | -43.6 | -46.4 | -42.9 | - | *** |
| North America 3 | $T_{warm}$ | 0.1 | 0.3 | -0.4 | 0.0 | -1.7 | -1.0 | +/- | |
| | $T_{cold}$ | -0.7 | -0.6 | 1.3 | 0.6 | -1.5 | -0.7 | - | * |
| | $P_{warm}$ | -18.2 | -15.5 | 31.4 | 21.3 | -23.1 | -12.2 | - | * |
| | $P_{cold}$ | 50.3 | 52.9 | 5.2 | 8.1 | 39.6 | 47.4 | + | *** |
| South America 2 | $T_{warm}$ | -1.8 | -1.7 | -3.3 | -3.1 | -0.7 | -0.3 | - | *** |
| | $T_{cold}$ | -2.1 | -1.7 | -2.9 | -3.0 | -1.6 | -1.7 | - | *** |
| | $P_{warm}$ | 125.6 | 96.7 | -45.1 | 57.0 | 7.7 | 14.7 | + | ** |
| | $P_{cold}$ | -13.5 | 2.1 | -0.8 | -0.9 | -75.3 | -89.7 | - | ** |
| Indo-Pacific 1 | $T_{warm}$ | -1.0 | -1.2 | 0.7 | 0.3 | 0.5 | 0.0 | +/- | |
| | $T_{cold}$ | 0.6 | 0.7 | 0.4 | 0.1 | 0.2 | 0.0 | + | ** |
| | $P_{warm}$ | 30.6 | -16.8 | -155.9 | -151.5 | 16.7 | -16.0 | - | * |
| | $P_{cold}$ | -45.2 | -38.8 | -89.8 | -87.2 | -119.1 | -106.4 | - | *** |

**Reviewer comment: (15)** *Section 4 (conclusions): this conclusion is short and superficial. What is the origin of the ESM biases according to reconstruction at least for the main ones? Has this paper filled the initial objectives?*

**Response:** Thank you for your comment, we have revised the conclusion.

**Revised text:**

[revised manuscript text omitted]

---

## Author Comment (AC2)

**Global biome changes over the last 21,000 years inferred from model-data comparisons**

**Response to comments of Anonymous Referee #2**

**1. General comments**

**Reviewer comment: (1)** *The paper of Li et al. uses a global set of pollen data to reconstruct megabiomes since the last ice age. This reconstruction is then compared with biomized outputs from Earth System Models in order to evaluate the fidelity of the model simulations. The biome reconstructions are quite interesting, and I appreciate the large fossil datasets compiled by the authors, this is huge effort to put together.*

**Response:** We appreciate the positive feedback on our work and the recognition of the effort involved in compiling the global pollen dataset for megabiome reconstructions. While we included a model-data comparison as an example study, the primary aim of our research is to present a new biomization dataset that offers the possibility to evaluate Earth System Models (ESMs). We have clarified this in our revised section "Summary and Conclusions." (Please see the next response)

**2. Specific Comments**

**Reviewer comment: (1)** *However, I have some comments about the methods as well as about the scientific contribution this makes through the interpretation of the data-model comparison. In particular, I find the conclusions to be very general and technical, and I hope that the authors are able to make the impacts of their study clearer.*

**Response:** Thank you for your comment, we have revised the Conclusions.

**Revised text:** (in red)

"This study presents a global megabiome reconstruction for 43 timeslices at 500-year intervals over the past 21,000 years, based on the most extensive taxonomically and temporally standardized fossil pollen dataset. The dataset's reliability is supported by a high agreement (~80%) with modern potential natural vegetation, and its general consistency with the paleosimulation ensemble further underscores its robustness for exploring past biome dynamics. With its high temporal and spatial coverage, it offers an unprecedented resource, not only for exploring long-term vegetation dynamics and their drivers, but also for diverse research contexts, including paleoclimate, biodiversity, and land-use studies. Furthermore, the dataset supports the evaluation of ESM-based paleo-megabiome simulations and offer insights for identifying potential biases in climate models. Its consistent

structure and broad applicability allow us to advance our integrative understanding of past, present, and future Earth system dynamics."

**Reviewer comment: (2)** *In my comment above, I say that I find the interpretation of the data-model comparison to be overly general. This is already observed in the abstract. The abstract ends with the statement: To some extent, these mismatches could be attributed to systematic model biases in the simulated climate, as well as to the different plant representations and low taxonomic resolution of pollen in the reconstructions.*

*I find this to be so general that it makes it really hard for the reader to glean any nuance the help us understand specific insights about model bias or data issues.*

*I suggest that the authors place more emphasis on the actual biome reconstruction from the pollen (which takes up more of the discussion, but is not much emphasized in the abstract) and take look deeper into the sources of mismatch to leave the reader with some key takeaways that relate directly to their stated goal (goal in abstract: to evaluate the paleosimulations from ESMs).*

**Response:** We have revised the Abstract based on your comments as follows:

**Revised text:**

"We present a global megabiome reconstruction for 43 timeslices at 500-year intervals throughout the last 21,000 years based on an updated and thus currently most extensive global taxonomically and temporally standardized fossil pollen dataset of 3,455 records. The evaluation with modern potential natural vegetation distributions yields an agreement of ~80%, suggesting a high reliability of the pollen-based megabiome reconstruction. With its high temporal and spatial resolution, this reconstruction is a robust dataset for various applications, such as the evaluation of paleo-simulations from Earth System Models (ESMs).

We compare the reconstruction with an ensemble of six biomized simulations based on transient vegetation simulations performed by ESMs. The global spatiotemporal patterns of megabiomes estimated by the simulation ensemble and reconstructions are generally consistent. That is, there has been a global shift from open glacial non-forest megabiomes to Holocene forest megabiomes since the Last Glacial Maximum (LGM), in line with the general climate warming trend and continental ice-sheet retreat. The shift to a global megabiome distribution generally similar to today's took place during the early Holocene. We also found that enhanced anthropogenic disturbances since the Late Holocene have not altered broad-scale megabiome patterns. However, certain data-model deviations are evident in specific regions and periods, which could be attributed to systematic climate biases in ESMs or biases in the pollen-based biomization method. For example, at a global scale over the last 21,000 years, the largest deviations between reconstructions and simulation ensembles are observed during the LGM and early deglaciation periods. These discrepancies are probably attributed to systematic summer cold biases in the ESMs that lead to an overestimation of tundra in periglacial regions. Additionally, steppes are overrepresented on the Tibetan Plateau in the reconstruction, because steppes share dominant characteristic taxa with tundra and are preferential allocated due to fewer plant functional types (PFTs) compared to tundra. Moderate deviations during

the Holocene mainly occur in non-forest megabiomes in the Mediterranean and North Africa, with increasing discrepancies over time. These deviations may result from the underestimation of woody PFT cover in simulations due to systematic biases, such as overly warm summers with dry winters in the Mediterranean, and the overrepresentation of woody taxa in reconstructions, misclassifying deserts as savanna in North Africa. On the whole, our reconstructions are suitable for the evaluation of ESM-based paleo-megabiome simulations, as well as providing clues for improving systematic model biases.**”**

**Reviewer comment: (3)** *This continues in the discussion with some very general statements about the sources of uncertainty in the comparison such as: We assume that the simulations used in this study share this rather common problem of a cold bias in boreal latitudes, resulting in the overestimation of tundra in simulations. There is no reference here to support the assumption and no sign that this potential bias was evaluated for the simulations used in the study.*

**Response:** We do not have pollen-independent climate reconstructions. So we cannot really say when and why the models have a climate that is too different, causing differences in vegetation compared to the reconstructions. There are some other reconstructions available, but it is not always clear which proxy records which climate variable, and deciding this is beyond our expertise, and a fair comparison of the past model climate with the reconstructions would be a huge effort and beyond the scope of this paper. Consequently, we limited the evaluation of the simulated climate to a comparison of the 0ka BP time-slice with modern observed climate data, which we added in the revised text.

**New text in Method:**

[revised manuscript text omitted]

**New figure (Appendix A3):**

[Figure]

[Figure]

**Figure A3. Differences in bioclimatic variables between ESM-based simulations at 0 cal. ka BP and observations.** The bioclimatic variables include **(a)** mean temperature of warmest quarter ($T_{warm}$), **(b)** mean temperature of coldest quarter ($T_{cold}$), **(c)** precipitation of warmest quarter ($P_{warm}$), and **(d)** precipitation of coldest quarter ($P_{cold}$). Notable biases include overestimated temperatures ($T_{warm}$ and $T_{cold}$) in highlands (e.g., Rockies, Andes, Tibetan Plateau), excessively dry Mediterranean summers ($P_{warm}$), and colder-than-observed conditions in Alaska ($T_{warm}$ and $T_{cold}$).

**Reviewer comment: (4)** *Another point I would add here is that the authors acknowledge the limitation of their modern validation exercise in incorporating human land use impacts to ecosystems when comparing with models that don't include such impacts. Please expand on how this might also impact data-model comparisons of paleo-simulations.*

**Response:** Thank you for your comment. We acknowledge the limitation of our modern validation exercise in not fully accounting for human land-use impacts, as the ESMs used in our comparisons do not include anthropogenic modifications. This has been clarified in the revised text, noting that while localized human activities may influence vegetation patterns, their impact on broad spatial and long-term paleo-simulations appears limited.

**Revised text:**

"Anthropogenic modification of pollen assemblages has, to some extent, contributed to mismatches in forested areas. For example, incorrectly reconstructed grasslands and dry shrublands (STEP) in North China may reflect intensive land use (e.g., deforestation).

[...]

Mid-Holocene to Late Holocene (represented by the timeslices 6 and 3 cal. ka BP): The spatial patterns of megabiome distributions during this period are only slightly different from those of the early Holocene. TRFO, for example, expanded in Mesoamerican reconstructions and simulations. It is also worth noting that the forest megabiomes have not obviously shifted since the Late Holocene, as revealed by both reconstructions and simulations. Given that the simulated vegetation was in a quasi-equilibrium with the climate and unaffected by humans, this implies a relatively stable climate in that period. Therefore, we propose that enhanced anthropogenic disturbances over this time period did not promote forest degradation at board spatial scale, and that biomization is robust as regards as this disturbance (Prentice et al., 1996; Gotanda et al., 2008)"

**Reviewer comment: (5)** *What about the impact of fire, I assume this is not included in ESMs? Could the lack of these processes in the models result in mismatch?*

**Response:** Thank you for your comment. All models used in this study include a fire module, which has been clarified in the revised text:

**Revised text:**

"The dynamic vegetation in all models is represented by different sets of plant functional types (PFTs) that can coexist in the grid-cells. The occurrence of each PFT is constrained by fixed temperature thresholds, and the dynamics of PFT cover fraction are depends for instance on the moisture availability and plant requirements. The fraction of PFTs is furthermore reduced by disturbances such as fire, windthrow, natural mortality, which are already coupled in the dynamic vegetation module (Dallmeyer et al., 2022). For instance, fire disturbances regularly decrease the tree and shrub PFT cover fractions, while promoting herbaceous PFTs expansion (Reick et al., 2021; Burton et al., 2019)."

**3. Minor comments**

**Reviewer comment: (1)** *Abstract: line 31 I don't understand term: global spatial megabiome. Does this mean the megabiomes of any particular time slice? I think there should be a way to simplify this.*

**Response:** We have revised the text as follows:

**Revised text:** (in red)

"The shift to a global megabiome distribution pattern similar to today's occurred during the early Holocene."

**Reviewer comment: (2)** *line 79: "8 of our own new records" there are no references for these.*

**Response:** We have included an overview table in Supplementary Data 1, which provides site metadata and lists the references for all records.

**Revised text:**

"Also, 52 records from the ACER 1.0 database (https://doi.org/10.1594/PANGAEA.870867; Sánchez Goñi et al., 2017), 177 records from the Chinese fossil pollen dataset (Zhou et al., 2023; Cao et al., 2022), and 8 of our own new records (AWI; for a detailed description see Supplementary Data 1, https://owncloud.gwdg.de/index.php/s/ijMPmsrahKFeY3Q/download) were included."

**Reviewer comment: (3)** *line 71: 3691 pollen records are in this compilation, but how many are included in the analysis after data filtration and quality control? This question applies to the numbers of records in Table 1 as well.*

**Response:** We have revised the text and updated Table 1 as follows:

**Revised text:**

"We converted pollen data from LegacyPollen 2.0 into megabiomes using the biomization method of Prentice et al. (1996). We only analyzed records over the last 21,000 years, resulting in a final megabiome dataset of 55,868 timeslices at 500-year intervals from 3,455 records (Supplementary Data 1 and Data 4; https://owncloud.gwdg.de/index.php/s/ijMPmsrahKFeY3Q/download)."

**Revised table:**

**Table 1:** Overview of the number of pollen records, pollen taxa, plant functional types (PFTs), and megabiomes used in the biomization procedures, along with references to biomization schemes by continent. The lists of taxa-PFTs and PFTs-megabiome assignments are available in Supplementary Data 2 and Data 3.

| Continent | Pollen records | Taxa | PFTs | Megabiomes | References |
|---|---|---|---|---|---|
| Europe | 1,359 | 243 | 41 | 7 | Ni et al. (2014) |
| | | | | | Binney et al. (2017) |
| | | | | | Marinova et al. (2018) |
| | | | | | Cao et al. (2019) |
| Asia | 636 | 424 | 49 | 8 | Chen et al. (2010) |
| | | | | | Ni et al. (2014) |
| | | | | | Binney et al. (2017) |
| | | | | | Tian et al. (2018) |
| | | | | | Cao et al. (2019) |
| North America | 1,078 | 393 | 47 | 8 | Thompson and Anderson (2000) |
| | | | | | Ortega-Rosas et al. (2008) |
| | | | | | Bigelow et al. (2003) |
| | | | | | Ni et al. (2014) |
| | | | | | Cao et al. (2019) |
| Africa | 145 | 556 | 8 | 6 | Vincens et al. (2006) |
| | | | | | Lézine et al. (2009) |
| Indo-Pacific | 60 | 429 | 22 | 8 | Pickett et al. (2004) |
| South America | 177 | 576 | 19 | 8 | Marchant et al. (2001 & 2009) |
| Total | 3,455 | 1,447 | 98 | 8 | |

**Reviewer comment: (4)** *line 82-83: following the previous question, I know this paper which provides recommendations for data best practices, but it doesn't specify specific practices for any study. Could you please tell us specifically how you filtered data? How many dates were required for age models, how were age models generated, how many pollen samples or counts were requires, etc?*

**Response:** We have revised the text according to your suggestion as follows:

**Revised text:**

"To improve comparability between pollen records as well as data quality, we followed the practices recommended by Flantua et al. (2023) for large-scale paleoecological data synthesis when updating the dataset.

Specifically, the following key steps were involved: first, metadata of pollen records from different data sources were examined to avoid duplicate inclusion; second, age-depth models were re-estimated for each record (≥ 2 radiocarbon dates) using the Bayesian framework implemented in Bacon (Blaauw and Christen, 2011; for a detailed description see Li et al., 2022); third, pollen morphotypes were harmonized to reduce the effect of taxonomic uncertainty and nomenclatural complexity, i.e. woody taxa and major herbaceous taxa have been harmonized to genus level and other herbaceous taxa to family level (for a detailed description see Herzschuh et al., 2022)."

**Reviewer comment: (5)** *Line 90: great that dois for specific datasets were included! This is great, helps attribute credit to individual record generators!*

**Response:** We appreciate your recognition of the inclusion of DOIs for specific datasets.

**Reviewer comment: (6)** *Line 110-111 "Larix and Pinus were multiplied by factors of 15 and 0.5" This is probably sensible for NA and Europe, but what about other overrepresented taxa in other regions?*

**Response:** We now acknowledge that the lack of calibration for taxa other than Larix and Pinus somewhat limits the accuracy of reconstructions in certain regions. However, we consider our global-scale, pollen-based megabiome reconstructions reliable because record-by-record comparisons of reconstructed megabiomes at timeslice 0 cal. ka BP from 2,232 available records with modern potential megabiomes indicate an 80.2% agreement.

**Revised text:**

"The different pollen representation (including production, dispersion, and preservation) of plant taxa is the principal reason for inadequate separation of forest and open landscape ecotones. For example, the high pollen productivity of key taxa (such as *Artemisia*; Xu et al., 2014) has resulted in an overestimation of grasslands and dry shrublands (STEP) in the East Asian summer monsoon northern marginal zone and the Great Plains of North America. However, studies on pollen productivity and dispersal ability to date are mostly limited to a few taxa in north-central Europe and China (Wieczorek and Herzschuh, 2020), which limits large-scale calibration of pollen representation."

**Reviewer comment: (7)** *Line 121: "Furthermore, the assignment of pollen taxa to megabiomes and biomization routines were performed independently for each continent." Is there specific information on the differences for each continent in that supplementary material or somewhere else? This is not clear to me. For example, different harmonization schemes have been published for different geographic areas, but I don't see a reference to this or other geographically specific procedures.*

**Response:** Yes, we have included the taxa-to-PFT-to-megabiome assignment scheme for each continent in the supplementary materials. We have clarified this point in the revised text:

**Revised text:**

"Furthermore, the assignment of pollen taxa to megabiomes and biomization routines were performed independently for each continent (Table 1; Supplementary Data 2 and Data 3, https://owncloud.gwdg.de/index.php/s/ijMPmsrahKFeY3Q/download)."

**Reviewer comment: (8)** *Line 159: regarding the tool of Dallmeyer et al., 2021, please provide a few details about how this works.*

**Response:** We have added details about how the tool of Dallmeyer et al., 2021 works. Please refer to the revised text below:

**Revised text:**

"The PFTs distributions are converted into the same eight megabiomes used in the reconstructions by applying the tool of Dallmeyer et al. 2019. This tool converts the simulated PFT distributions based on assumptions of the minimum PFT cover fractions that is needed for the assignment of steppe/tundra or forest biomes and bioclimatic constraints derived from 2 m temperature distributions to distinguish different forest biomes (for a detailed description see Dallmeyer et al. 2019). These constraints largely adhere to the limitation rules used in the classical biome models such as BIOME4 (Kaplan et al., 2003). This two-way approach of using temperature constraints and PFT cover fractions is needed to assure the general application of the tool to all standard ESMs, that sometimes only calculate two different PFTs (herbaceous and trees)."

**References:**

[revised manuscript text omitted]

---

## Author Response (AR1)

**Global biome changes over the last 21,000 years inferred from model-data comparisons**

**Response to comments of Referee #1 Guiot Joel**

**1. General comments**

**Reviewer comment: (1)** *Authors present a new global dataset of 3691 pollen records that are transformed into biomes using the well-established method of Prentice et al (1996). The dataset covers the whole globe with some gaps in Africa, southern Asia and Australia, where the data are not available.*

**Response:** We acknowledge the limited availability of data in certain regions and have clarified this in the text.

**Text: Line 87-88:** "While there are geographical gaps in pollen record coverage, particularly in the Southern Hemisphere, the dataset LegacyPollen 2.0 covers the world's main vegetation and climate zones."

**Reviewer comment: (2)** *They work at the level of 8 mega biomes which represent raw pattern of global vegetation. Indeed, it is difficult to work with finer biomes at a global scale (even if Prentice at al 2000 did for two time slices). The biomes reconstructed from 21 ka BP to present at 500 years' time steps are compared with equivalent biomes simulated by three models (and two simulations each).*

**Response:** Thank you for your comment. We have revised the text to clarify the classification of megabiomes and its application in data-model comparison as follows:

**Revised text**: (in red)

**Line 110-117:** "The PFTs were then assigned to megabiomes, representing the raw pattern of global vegetation rather than the finer biome categories commonly used in standard biomization studies (Dallmeyer et al., 2019). These megabiomes include tropical forest (TRFO), subtropical forest (WTFO), temperate forest (TEFO), boreal forest (BOFO), (warm-) savanna and dry woodland (SAVA), grassland and dry shrubland (STEP), (warm) desert (DESE), tundra and polar desert (TUND). These categories were also applied to biomize Earth System Model results, which generally use different types and numbers of PFTs to represent global vegetation, enabling direct data-model comparisons and evaluations (Dallmeyer et al., 2019)."

**Reviewer comment: (3)** *The main problem of the paper can be easily detected in the abstract (which reflect correctly the full paper). The paper is technical and too descriptive.*

**Response:** Thank you for your comment. We have revised the abstract to better emphasize the key findings and broader implications while reducing technical descriptions.

**Revised text:**

**Line 23-47:** "We present a global megabiome reconstruction for 43 timeslices at 500-year intervals throughout the last 21,000 years,based on an updated and thus currently most extensive global taxonomically and temporally standardized fossil pollen dataset of 3,455 records. The evaluation with modern potential natural vegetation distributions yields an agreement of ~80%, suggesting a high reliability of the pollen-based megabiome reconstruction.

We compare the reconstruction with an ensemble of six biomized simulations derived from transient Earth System Models (ESMs). Overall, the global spatiotemporal patterns of megabiomes estimated by both, the simulation ensemble and reconstructions, are generally consistent. Specifically, they reveal a global shift from open glacial non-forest megabiomes to Holocene forest megabiomes since the Last Glacial Maximum (LGM), in line with the general climate warming trend and continental ice-sheet retreat. The shift to a global megabiome distribution generally similar to today's took place during the early Holocene; Furthermore, the reconstructions reveal that enhanced anthropogenic disturbances since the Late Holocene have not altered broad-scale megabiome patterns.

However, certain data-model deviations are evident in specific regions and periods, which could be attributed to systematic climate biases in ESMs or biases in the pollen-based biomization method. For example, at a global scale over the last 21,000 years, the largest deviations between reconstructions and simulation ensemble are observed during the LGM and early deglaciation periods. These discrepancies are probably attributed to the ESM systematic summer cold biases that overestimate tundra in periglacial regions, as well as the challenging identification of steppes and tundra from the Tibetan Plateau pollen records. Moderate deviations during the Holocene mainly occur in non-forest megabiomes in the Mediterranean and North Africa, with increasing discrepancies over time. These deviations may result from the underestimation of woody PFT cover in simulations due to systematic biases, such as overly warm summers with dry winters in the Mediterranean, and the overrepresentation of woody taxa in reconstructions, misclassifying deserts as savanna in North Africa.

Overall, our reconstruction, with its relatively high temporal and spatial resolution, serves as a robust dataset for evaluating ESM-based paleo-megabiome simulations, as well as providing potential clues for improving systematic model biases."

**Reviewer comment: (4)** *Biomes are reconstructed, simulated with ESM and compared, some biases of the ESM simulations are pointed out and that is all. There are no general messages on the evolution of the vegetation through the 21 millennia.*

**Response:** We have described the global vegetation evolution over the past 21,000 years in section 3.1 as follows:

**Text: Line 350-414:** "We present a global assessment of megabiome dynamics and distributions derived from

pollen-based reconstructions and ESM-based simulations over the last 21,000 years, with a temporal resolution of 500 years. Overall, there has been a global shift from open glacial non-forest megabiomes to Holocene forest megabiomes since the LGM (Fig. 2), in line with the general climate warming trend and continental ice-sheet retreat (Fig. 3):

LGM (represented by the timeslice 21 cal. ka BP): TUND and BOFO dominate the high latitudes and periglacial areas (similar to Prentice et al., 2000 and Davis et al., 2024), whereas the relatively warm forest megabiomes (e.g., WTFO and TEFO) are distributed at lower latitudes than present-day, in response to cold and dry climates. [...]**"**

**Reviewer comment: (5)** *The ESM biases are not discussed by analysing which climatic variables are responsible of them.*

**Response:** Thank you for your comment. We performed further analyses to assign the ESM biases to certain climatic variables. Due to the lack of pollen-independent climate reconstructions at broad spatial and long temporal scales, we cannot determine when and why models exhibit climate biases. Consequently, we limited our comparison to modern observed climate data in the revised text.

**New text in Method:**

**Line 186-199:** "Modern observational climate data provide a crucial foundation for the assessment of climate simulations. The Climatic Research Unit gridded Time Series (CRU TS hereafter), version 4.08, is a widely used modern observational climate dataset covering all land domains of the world except Antarctica (spatial resolution: ~0.5°x0.5° on a Gaussian grid; Harris et al., 2020). The CRU TS dataset is interpolated from extensive networks of weather station observations and provides monthly temperature and precipitation data from 1901-2023 C.E. However, the early records (i.e., < 1930 C.E.) of this dataset may have high uncertainty due to sparser observation networks (Duan et al., 2024), and the late records (i.e., > 1970 C.E.) is strongly influenced by anthropogenic $CO_2$ increases (Cheng et al., 2022). We, therefore, selected monthly climatological means from 1931-1970 to generate more biologically meaningful bioclimatic variables for evaluating climate simulations at 0 cal. ka BP (O'Donnell and Ignizio, 2012; Supplementary Data 7). These bioclimatic variables represent extreme or limiting environmental factors, namely, mean temperature of warmest quarter (Twarm), mean temperature of coldest quarter (Tcold), precipitation of warmest quarter (Pwarm), and precipitation of coldest quarter (Pcold). Temperature is given in degrees Celsius (°C), precipitation in millimeters (mm), and a quarter is a period of three consecutive months (1/4 of the year)."

**Revised text in Results and Discussion:**

**Line 327-349:** "The agreement between modern potential megabiomes and simulated megabiomes at timeslice 0 cal. ka BP is higher for the ESM-representative megabiome (cf. Sect. 2.3) than for individual ESM-based simulations (64.1% vs. 20.0–60.2%; Table 3). As a result, the ESM-representative megabiome depicts more reliable patterns of megabiome dynamics and distribution than individual simulations, with higher agreement

[revised manuscript text omitted]

**Reviewer comment: (6)** *Why mediterranean biomes of the Holocene are simulated as steppes whereas data indicate TEDE?*

**Response:** All models consistently simulate steppe (STEP) rather than temperate forest (TEDE) in the Mediterranean region during the Holocene, likely due to systematic biases in the simulated climate, as described in the revised text.

**Revised text:**

**Line 368-373:** "The Mediterranean region has hot dry summers and mild wet winters. Modeling studies report systematic model biases of hotter summers and drier winters in this region (García-Herrera and Barriopedro, 2018). A comparison with modern data shows similar climate biases in the simulations which may indicate similar systematic biases in the past. This would explain the underrepresentation of the cover fraction of woody PFTs in the simulations (Fig. A3a, d)."

**Reviewer comment: (7)** *There are also biases in the reconstructed biomes. As an example, the glacial mediterranean biomes are sometimes reconstructed as TEDO while previous reconstructions (Elenga et al, 2000; Prentice et al, 2000) reconstructed STEP.*

**Response:** Thank you for your comment. We have added more comparisons in the revised text to clarify the differences between our reconstructions and previous studies.

**Revised text:** (in red)

**Line 371-384:** "LGM (represented by the timeslice 21 cal. ka BP): TUND and BOFO dominate the high latitudes and periglacial areas (similar to Prentice et al., 2000), whereas the relatively warm forest megabiomes (e.g., WTFO and TEFO) are distributed at lower latitudes than present-day, in response to cold and dry climates (Nolan et al., 2018). However, the ESM-representative megabiome (simulations hereafter in this Sect.) reveals more non-forest megabiomes (such as TUND and STEP) in periglacial areas of North America (e.g., Alaska and the Rocky Mountains) and northern Asia (e.g., northeastern Siberia), as well as in the Mediterranean regions, as compared

to the reconstructions. Although previous pollen-based biomization studies with different biomization schemes have reported ESM-like results (such as Binney et al., 2017 and Cao et al., 2019 in periglacial areas; Elenga et al., 2000 and Prentice et al., 2000 in the Mediterranean regions), assessments of modern megabiome distributions suggest that these studies overestimated the occurrence of non-forest megabiomes in these regions. A recent pollen-based forest cover reconstruction by Davis et al. (2024) indicates more forest than previously suggested by biome reconstructions in these regions during the LGM, which aligns with our results. Furthermore, STEP occurred in central Asia in the reconstructions rather than TUND in the simulations, and TRFO and SAVA appeared in tropical South America and Africa in the reconstructions rather than WTFO in the simulations.**"**

**Reviewer comment: (8)** *I have the feeling that for the regions I know better, this paper does a poorer reconstruction than previous attempts. If not, this should be argued.*

**Response:** Thank you for your comment. We have added regional comparisons in the revised text to further clarify the reliability of our reconstruction.

**Revised text:**

**Line 261-279: "**We consider global-scale, pollen-based megabiome reconstructions to be reliable, as record-by-record comparisons of reconstructed megabiomes at timeslice 0 cal. ka BP from 2,232 available records with modern potential megabiomes indicate an 80.2% agreement (Table 3). This consistency exceeds that reported in previous large-scale biomization studies validated against modern biome distributions, such as the 53% agreement in Arctic high-latitudes (>55°N) by Bigelow et al. (2003). We attribute this high agreement not only to the high quality of the pollen dataset, particularly in terms of taxonomic and temporal harmonization, but also to the biomization method that employs updated and harmonized schemes assigning pollen taxa to plant functional types to megabiomes. Additionally, our reconstruction was conducted at the megabiome level, a coarser classification than typical biomes, which somewhat reduces mismatches between geographically adjacent biomes. For instance, the biomes of temperate deciduous forest and cool mixed forest are often intermingled in Binney et al. (2017), whereas at the megabiome level, both are classified as temperate forests, eliminating this discrepancy. Although some regional-scale biomization studies achieve even higher agreement with modern biome distributions, such as the 97.5% accuracy in the Congo Basin reported by Lebamba et al. (2009), these studies typically rely on more localized datasets with tailored taxa-PFT-biome schemes. Moreover, not all megabiomes were reconstructed with the same level of accuracy in our study, such as the TUND and STEP exhibit only ~50% agreement (Table 3), which similar to previous biomization studies. Overall, we argue that the data quality as well as the higher spatial and temporal coverage compared to previous biomization studies (Bigelow et al., 2003; Marinova et al., 2018) make our pollen-based megabiome reconstruction a robust dataset for various applications, such as global-scale evaluation of paleo-simulations from Earth System Models (ESMs).**"**

**Reviewer comment: (9)** *In conclusion, the paper has potentialities, but it needs more work to be a good contribution to the discipline.*

**Response:** We sincerely appreciate the reviewer's thoughtful feedback and recognition of the potential of our work. We have carefully addressed the issues raised in the review, please refer to the previous responses.

**2. Specific comments**

**Reviewer comment: (1)** *Section 2.3: how are interpolated the biomes in sites from the grid of the ESM (the dots of Fig 1B)?*

**Response:** Thank you for your comment. We have revised the text in Section 2.3:

**Revised text:** (in red)

**Line 179-181:** "We assigned the simulated megabiome data taken from the grid-cells where the records are located to each record, and we only considered locations and timeslices for which reconstructions are available (Supplementary Data 4-6)."

**Reviewer comment: (2)** *230-233: I disagree that there are no systematic mismatch: there are in the Med region, in the subarctic one … Note also that TUND and STEP have low agreement (50%)*

**Response:** Thank you for your comment. The sentence has been removed, and the lower agreement of TUND and STEP has been clarified in the revised text.

**Revised text:**

**Line 294-309:** "The low taxonomic resolution could also cause mismatches between neighboring forest megabiomes, as well as between tundra (TUND) and grassland (STEP). Woody taxa have been harmonized to the genus level rather than the species level, while herbaceous taxa are generally harmonized to the family level, except for common taxa like *Artemisia*, *Thalictrum*, and *Rumex*. This reduces the ecological information available for PFT assignment (Chen et al., 2010).

[...]

Similarly, TUND may have been misrepresented as STEP on the Tibetan Plateau. This misrepresentation can be attributed to their share of dominant characteristic taxa within Poaceae and Cyperaceae. However, STEP is defined by fewer PFTs and therefore preferentially allocated to samples. In contrast, woody PFTs are generally not defined in STEP, leading to a potential misallocation to TUND rather than STEP in cases of woody pollen grain occurrences (from long-distance transportation or local existence) in open landscape samples (Marinova et al., 2018; Chen et al., 2010), such as mismatches in southern Europe."

**Reviewer comment: (3)** *233-235: I think that the good agreement comes to the fact that the comparison is restricted to 8 megabiomes. In the previous papers, finer biomes are considered and often the mismatch is between climatically neighbor biomes (this is less possible with megabiomes).*

**Response:** We agree that the good agreement is partly due to being limited to 8 megabiomes. We have revised the text accordingly.

**Revised text:**

**Line 261-279:** "We consider global-scale, pollen-based megabiome reconstructions to be reliable, as record-by-record comparisons of reconstructed megabiomes at timeslice 0 cal. ka BP from 2,232 available records with modern potential megabiomes indicate an 80.2% agreement (Table 3). This consistency exceeds that reported in previous large-scale biomization studies validated against modern biome distributions, such as the 53% agreement in Arctic high-latitudes (>55°N) by Bigelow et al. (2003). We attribute this high agreement not only to the high quality of the pollen dataset, particularly in terms of taxonomic and temporal harmonization, but also to the biomization method that employs updated and harmonized schemes assigning pollen taxa to plant functional types to megabiomes. Additionally, our reconstruction was conducted at the megabiome level, a coarser classification than typical biomes, which somewhat reduces mismatches between geographically adjacent biomes. For instance, the biomes of temperate deciduous forest and cool mixed forest are often intermingled in Binney et al. (2017), whereas at the megabiome level, both are classified as temperate forests, eliminating this discrepancy. Although some regional-scale biomization studies achieve even higher agreement with modern biome distributions, such as the 97.5% accuracy in the Congo Basin reported by Lebamba et al. (2009), these studies typically rely on more localized datasets with tailored taxa-PFT-biome schemes. Moreover, not all megabiomes were reconstructed with the same level of accuracy in our study, such as the TUND and STEP exhibit only ~50% agreement (Table 3), which similar to previous biomization studies. Overall, we argue that the data quality as well as the higher spatial and temporal coverage compared to previous biomization studies (Bigelow et al., 2003; Marinova et al., 2018) make our pollen-based megabiome reconstruction a robust dataset for various applications, such as global-scale evaluation of paleo-simulations from Earth System Models (ESMs)."

**Reviewer comment: (4)** *259-263: I do not understand this sentence.*

**Response:** We apologize for the lack of clarity in the original text. The sentence has been revised for better understanding as follows:

**Revised text:**

**Line 285-290:** "Similarly, TUND may have been misrepresented as STEP on the Tibetan Plateau. This misrepresentation can be attributed to their share of dominant characteristic taxa within Poaceae and Cyperaceae. However, STEP is defined by fewer PFTs and therefore preferentially allocated to samples. In contrast, woody PFTs are generally not defined in STEP, leading to a potential misallocation to TUND rather than STEP in cases

of woody pollen grain occurrences (from long-distance transportation or local existence) in open landscape samples (Marinova et al., 2018; Chen et al., 2010), such as mismatches in southern Europe."

**Reviewer comment: (5)** *Section 3.1.2: It is necessary to try to explain the discrepancies between simulated biomes and potential vegetation by over or under-simulation of some climate variables.*

**Response:** We have incorporated an additional evaluation using modern climate data as follows:

**Revised text:**

**Line 308-330:** "The agreement between modern potential megabiomes and simulated megabiomes at timeslice 0 cal. ka BP is higher for the ESM-representative megabiome (cf. Sect. 2.3) than for individual ESM-based simulations (64.1% vs. 20.0–60.2%; Table 3). As a result, the ESM-representative megabiome depicts more reliable patterns of megabiome dynamics and distribution than individual simulations, with higher agreement especially in Alaska, the Iberian Peninsula, the Alps, the Atlantic Coastal Plain of North America, and the southeastern United States (Fig. 1 and Fig. A2). However, there are still certain regions with low agreement, probably due to climatic biases. These include nearly all highlands (such as the central-southern Rocky Mountains, the central Andes, and the Tibetan Plateau) for which an overestimation of the temperature can be expected in the models due to a much lower orography than in reality caused by the smoothing in the coarse spatial resolution (3.75°x3.75° and 5°x5°) of the model grids (Fig. A3a–b). All models simulate non-forest megabiomes instead of forest in the Mediterranean region, which can be attributed to the models simulating a climate that is too seasonally dry, with, for example, too-warm summers and too-dry winters (Fig. A3a, d). The TRACE-21K simulation as well as the MPI-ESM simulations fail to reproduce the boreal forest (BOFO) in Alaska, which is then also reflected in the ESM-representative megabiomes. This failure is likely due to the simulated climate being too cold in this region, preventing the establishment of boreal forests under modeled conditions (Fig. A3a, d). Similar to the reconstructions, the transition zones between temperate forest (TEFO) and non-forest megabiomes, such as the East Asian summer monsoon margin, are regions with lower simulated megabiome agreement to the modern potential megabiome distribution. In North Africa, the models also tend to underestimate the northern extension of the grassland and dry shrubland (STEP) and incorrectly assign (warm) savanna and dry woodland (SAVA) records to tropical forest (TRFO). This is related to the biomization procedure for the model results that only relies on simulated vegetation cover fractions and simulated climate, whereas savannas are additionally determined by other ecological processes such as fire intensity and frequency (Dallmeyer et al., 2019) or grazing (van Langevelde et al., 2019)."

**Reviewer comment: (6)** *Figure 3: how is obtained the ice-sheet extension data from pollen?*

**Response:** Thank you for your comment. To clarify, the ice-sheet extension data in Figure 2 are not derived from pollen data. Instead, they are based on an ice-sheet ensemble set designed for fair comparisons among simulations.

**Revised Figure 2 caption:**

**Line 339-341:** "The ice sheets are shown at their maximum extent at timeslices synthesized for the ICE-5G (Peltier, 2004), ICE-6G (Peltier et al., 2015), and GLAC-1D (Tarasov et al., 2012) reconstructions."

**Reviewer comment: (7)** *320-331: Elenga et al 2000 and Prentice et al 2000 reconstructed steppes at 21 ka BP, which seems more realistic than tundra. Prentice et al 2000 is not cited despite the fact that they produced a full global reconstruction for the 21 ka BP period (and also mid-Holocene). It is a major paper that the authors cannot ignore. 325-332: the problem of assigning TUND to STEP and vice versa should also be discussed; previous papers reconstruct mainly STEP to the Med region and not TUND as here. There are also biases in the reconstructed biomes.*

**Response:** Thanks for your recommendation and we have included this reference. Our reconstruction may have misrepresented STEP as TUND, which has been discussed in the revised text:

**Revised text:**

**Line 275-290:** "(b) The low taxonomic resolution could also cause mismatches between neighboring forest megabiomes, as well as between tundra (TUND) and grassland (STEP). Woody taxa have been harmonized to the genus level rather than the species level, while herbaceous taxa are generally harmonized to the family level, except for common taxa like *Artemisia*, *Thalictrum*, and *Rumex*. This reduces the ecological information available for PFT assignment (Chen et al., 2010).

[...]

Similarly, TUND may have been misrepresented as STEP on the Tibetan Plateau. This misrepresentation can be attributed to their share of dominant characteristic taxa within Poaceae and Cyperaceae. However, STEP is defined by fewer PFTs and therefore preferentially allocated to samples. In contrast, woody PFTs are generally not defined in STEP, leading to a potential misallocation to TUND rather than STEP in cases of woody pollen grain occurrences (from long-distance transportation or local existence) in open landscape samples (Marinova et al., 2018; Chen et al., 2010), such as mismatches in southern Europe."

**Reviewer comment: (8)** *333-334: TUND seems more extended in Europe at 16-13ka than at 21ka, while the warming has already started, why?*

**Response:** Thank you for your comment. We have clarified this in the revised text:

**Revised text:**

**Line 366-370:** "Deglaciation (represented by the timeslices 16 and 13 cal. ka BP): Compared with the LGM, the extratropical megabiomes experienced a remarkable expansion to higher latitudes that coincided with the retreat of the continental ice sheets (Fig. 3). In particular, BOFO, TUND, and TEFO underwent a more extensive expansion compared to the other megabiomes in both our reconstructions and simulations; a result similar to previous biomization studies (such as Binney et al., 2017 and Cao et al., 2019 north of 30°N)."

**Reviewer comment: (9)** *342-353: Biomization starts to be realistic in the Holocene, much more than for the cold periods. But It is strange that there is no Mediterranean vegetation (WTFO) in the Med area during all the Holocene, as well for simulation as reconstruction. For simulation WTFO is replaced by STEP and for reconstruction, it is by TEFO.*

**Response: (a)** All models consistently simulate steppe (STEP) rather than subtropical forest (WTFO) in the Mediterranean region during all the Holocene, likely due to systematic biases in the simulated climate, as described in the revised text.

**Revised text:**

**Line 449-454:** "The Mediterranean region has hot dry summers and mild wet winters. Modeling studies report systematic model biases of hotter summers and drier winters in this region (García-Herrera and Barriopedro, 2018). A comparison with modern data shows similar climate biases in the simulations which may indicate similar systematic biases in the past. This would explain the underrepresentation of the cover fraction of woody PFTs in the simulations (Fig. A3a, d)."

**(b)** In our reconstruction, we may have misrepresented WTFO as TEFO, which has been clarified in the revised text.

**Revised text:**

**Line 275-285:** "The low taxonomic resolution could also cause mismatches between neighboring forest megabiomes, as well as between tundra (TUND) and grassland (STEP). Woody taxa have been harmonized to the genus level rather than the species level, while herbaceous taxa are generally harmonized to the family level, except for common taxa like *Artemisia*, *Thalictrum*, and *Rumex*. This reduces the ecological information available for PFT assignment (Chen et al., 2010). For instance, different species within *Pinus*, *Alnus*, *Fagus*, and *Betula* (Tian et al., 2018) have different bioclimatic controls, phenology, and life forms, but identification at the genus level results in them being shared by key PFTs in different forest megabiomes (e.g., WTFO vs. TEFO, TEFO vs. BOFO) when assigning taxa to PFTs. One of the typical areas in which this problem occurs is southern Scandinavia. Pollen grains from *Betula pendula* in temperate forests and *Betula pubescens* in boreal forests (Beck et al., 2016) in this region can only be identified to genus level, resulting in these two key species not being good indicators of temperate and boreal forests."

**(c)** However, the WFTO is not the dominant potential natural megabiome in the modern Mediterranean region. As shown in the figure below, which depicts the distribution of modern potential natural megabiomes aggregated from modern potential natural vegetation (Ramankutty and Foley, 1999; Ramankutty et al., 2010). It represents

the world's vegetation cover that had most likely existed for 1986–1995 C.E. in equilibrium with present-day climate and natural disturbance in the absence of human activities.

[Figure]

**Figure.** Spatial patterns of modern potential natural megabiome distributions.

**Reviewer comment: (10)** *356-361: I read on the maps the opposite to what is claimed: I see a forest degradation in the simulations (maps at right) not in the reconstructions (maps at left). In the whole Holocene, reconstructions show a constant TEFO, while simulation shows steppes in the second part of the Holocene, tending to show that there is a bias towards aridity in the models. Pollen reconstructions may sometimes be influenced by human deforestation, but it has been shown in previous papers that biomization is robust as regards as this disturbance.*

**Response:** We have revised this paragraph based on your suggestion.

**Revised text:**

**Line 388-395: "**Mid-Holocene to Late Holocene (represented by the timeslices 6 and 3 cal. ka BP): The spatial patterns of megabiome distributions during this period are only slightly different from those of the early Holocene. TRFO, for example, expanded in Mesoamerican reconstructions and simulations. It is also worth noting that the forest megabiomes have not obviously shifted since the Late Holocene, as revealed by both reconstructions and simulations. Given that the simulated vegetation was in a quasi-equilibrium with the climate and unaffected by humans, this implies a relatively stable climate in that period. Therefore, we propose that enhanced anthropogenic disturbances over this time period did not promote forest degradation at a broad spatial scale, and that biomization is robust regarding these disturbance (Prentice et al., 1996; Gotanda et al., 2008)."

**Reviewer comment: (11)** *Figure 4: It does not exist the possibility to computed significance levels for EDM?*

**Response:** Thank you for the comment. While it is technically feasible to compute significance levels for the EMD, we have chosen not to perform such calculations as they are not directly relevant to the goals of our study.

The Earth Mover's Distance (EMD) is designed to quantify the degree of mismatch between reconstructions and simulations by integrating uncertainties and weighted ecological and climatic distances. Its primary purpose is to provide a nuanced, distribution-based metric for evaluating spatiotemporal patterns of biome differences rather than testing for random variation or statistical significance.

Significance levels typically require assumptions about the underlying data distribution or the application of resampling techniques, such as bootstrapping. However, the weighting scheme we employed in our EMD calculations is highly context-specific, reflecting ecological and climatic gradients rather than stochastic variation. This specificity makes conventional significance testing less informative in our context.

Moreover, the core objective of our analysis is to assess the relative agreement between reconstructions and simulations across regions and timeslices, focusing on pattern visualization and identifying areas with stronger or weaker model-data agreement. Introducing significance levels would not substantially contribute to the interpretability or scientific value of our results, as the threshold would be inconsistent across different spatial and temporal contexts due to variations in data density and spatial autocorrelation.

To ensure methodological clarity, we emphasize the robustness of our approach by employing carefully defined affinity scores and weighting schemes that directly address vegetation-climate dynamics. These components have been selected to ensure that the EMD reflects biologically and climatically meaningful differences, aligning with the central aims of our study.

**Reviewer comment: (12)** *Caption of Fig.4: The sentence "The largest datamodel deviations occur during the LGM and early deglaciation periods" should not be put in the caption.*

**Response:** We've removed it as you suggested.

**Reviewer comment: (13)** *385-389: "the best data-model agreement occurs during the Bølling-Allerød interstadial (represented by the timeslice 14 cal. ka BP)": This appears to be true with the global EMD but in the regional EMD, it does not appear that 14ka EDM was minimum in any regions.*

**Response:** The global data-model EMD at each timeslice is derived from the median EMDs of clustered regions at that timeslice. While the global EMD indicates the best data-model agreement during the Bølling-Allerød interstadial (14 cal. ka BP), this does not necessarily mean that the EMD for this timeslice was the minimum in any specific region. Instead, the global pattern represents the aggregated dynamics across all regions, where regional variations may exhibit significant heterogeneity. This distinction reflects the synthesis approach used in our analysis and highlights the importance of considering both global and regional perspectives.

**Reviewer comment: (14)** *Section 3.3: it should be useful to summarize by a table the biases found in simulations and giving an interpretation of which climate variable is responsible of the biases*

**Response:** We do not have pollen-independent climate reconstructions. So we cannot really say when and why the models have a climate that is too different, causing differences in vegetation compared to the reconstructions. There are some other reconstructions available, but it is not always clear which proxy records which climate variable, and deciding this is beyond our expertise, and a fair comparison of the past model climate with the reconstructions would be a huge effort and beyond the scope of this paper. Therefore, we can only compare the modern observed climate with the models and add a new table to the appendix.

**New table:**

**Line 560-567: Table A1. The median difference in bioclimatic variables between ESM-based simulations at 0 cal. ka BP and observations by regions.** The regional clustering is shown in Figure 4b. Bioclimatic variables: $T_{warm}$ - mean temperature of warmest quarter, $T_{cold}$ - mean temperature of coldest quarter, $P_{warm}$ - precipitation of warmest quarter, and $P_{cold}$ - precipitation of coldest quarter. A positive sign in the simulation ensemble difference indicates that the number of simulations that overestimate the bioclimatic variable is greater than the number that underestimate it among the six simulations, while a negative sign indicates the opposite, and positive/negative signs indicate that they are equivalent. Confidence among the six simulations is indicated by one, two, and three asterisks for four, five, and six simulations sharing the same sign, respectively.

| Regions | Bioclimatic variables | MPI-ESM | | CLIMBER-X | | CCSM3 | | Simulation ensemble | |
|---|---|---|---|---|---|---|---|---|---|
| | | MPI-ESM_GLAC1D | MPI-ESM_ICE6G | CLIMBER-X_GLAC1D | CLIMBER-X_ICE6G | TRACE-21K-I_ICE5G | TRACE-21K-II_ICE5G | Difference | Confidence |
| Asia 1 | $T_{warm}$ | -2.4 | -2.3 | 3.1 | 3.6 | -10.0 | -5.5 | - | * |
| | $T_{cold}$ | 1.9 | 3.1 | 4.3 | 4.7 | -10.4 | -9.9 | + | * |
| | $P_{warm}$ | 36.3 | 26.6 | 43.9 | 52.2 | -25.9 | -1.1 | + | * |
| | $P_{cold}$ | 10.5 | 12.1 | 26.6 | 23.7 | -3.0 | 1.8 | + | ** |
| North America 1 | $T_{warm}$ | -0.4 | -0.4 | 0.8 | 0.7 | -8.2 | -5.4 | - | * |
| | $T_{cold}$ | 3.3 | 4.7 | 3.4 | 3.7 | -6.3 | -5.5 | + | * |
| | $P_{warm}$ | 81.9 | 55.4 | 24.7 | 29.7 | -12.7 | 12.4 | + | ** |
| | $P_{cold}$ | 31.6 | 30.7 | 64.8 | 58.8 | 18.7 | 20.7 | + | *** |
| Asia 2 | $T_{warm}$ | -3.8 | -3.6 | 1.7 | 2.0 | -8.2 | -4.1 | - | * |
| | $T_{cold}$ | 2.7 | 3.3 | 6.6 | 6.4 | -5.2 | -3.2 | + | * |
| | $P_{warm}$ | 34.7 | 19.0 | 14.2 | 38.3 | -71.3 | -53.3 | + | * |
| | $P_{cold}$ | 29.3 | 25.9 | -14.2 | -6.7 | 6.7 | 13.6 | + | * |
| Asia 4 | $T_{warm}$ | 0.7 | 1.1 | -0.8 | 0.6 | 2.0 | 2.4 | + | ** |
| | $T_{cold}$ | 0.4 | 0.5 | -0.3 | 0.4 | -2.3 | -2.5 | +/- | |
| | $P_{warm}$ | 0.7 | -25.3 | -33.0 | -33.5 | -4.3 | -0.6 | - | ** |
| | $P_{cold}$ | 51.2 | 49.5 | -0.4 | 1.1 | 70.3 | 79.4 | + | ** |
| Africa 1 | $T_{warm}$ | -0.6 | -0.2 | 0.8 | -0.5 | -1.8 | -1.0 | - | ** |
| | $T_{cold}$ | -0.8 | -1.2 | 1.9 | 0.8 | -1.5 | -1.0 | - | * |
| | $P_{warm}$ | -13.6 | -13.6 | 23.7 | 24.3 | 2.8 | 5.4 | + | * |
| | $P_{cold}$ | -0.5 | -0.5 | 13.2 | 14.0 | -2.3 | -2.4 | - | * |
| Europe 2 | $T_{warm}$ | 0.9 | 0.5 | 0.0 | -0.9 | -2.8 | -1.0 | - | * |
| | $T_{cold}$ | 1.1 | 1.4 | 2.0 | 1.3 | -0.5 | 0.8 | + | ** |
| | $P_{warm}$ | -17.9 | -18.1 | 38.8 | 36.4 | -5.6 | -4.6 | - | * |
| | $P_{cold}$ | -26.2 | -52.3 | -44.0 | -43.6 | -46.4 | -42.9 | - | *** |
| North America 3 | $T_{warm}$ | 0.1 | 0.3 | -0.4 | 0.0 | -1.7 | -1.0 | +/- | |
| | $T_{cold}$ | -0.7 | -0.6 | 1.3 | 0.6 | -1.5 | -0.7 | - | * |
| | $P_{warm}$ | -18.2 | -15.5 | 31.4 | 21.3 | -23.1 | -12.2 | - | * |
| | $P_{cold}$ | 50.3 | 52.9 | 5.2 | 8.1 | 39.6 | 47.4 | + | *** |
| South America 2 | $T_{warm}$ | -1.8 | -1.7 | -3.3 | -3.1 | -0.7 | -0.3 | - | *** |
| | $T_{cold}$ | -2.1 | -1.7 | -2.9 | -3.0 | -1.6 | -1.7 | - | *** |
| | $P_{warm}$ | 125.6 | 96.7 | -45.1 | 57.0 | 7.7 | 14.7 | + | ** |
| | $P_{cold}$ | -13.5 | 2.1 | -0.8 | -0.9 | -75.3 | -89.7 | - | ** |
| Indo-Pacific 1 | $T_{warm}$ | -1.0 | -1.2 | 0.7 | 0.3 | 0.5 | 0.0 | +/- | |
| | $T_{cold}$ | 0.6 | 0.7 | 0.4 | 0.1 | 0.2 | 0.0 | + | ** |
| | $P_{warm}$ | 30.6 | -16.8 | -155.9 | -151.5 | 16.7 | -16.0 | - | * |
| | $P_{cold}$ | -45.2 | -38.8 | -89.8 | -87.2 | -119.1 | -106.4 | - | *** |

**Reviewer comment: (15)** *Section 4 (conclusions): this conclusion is short and superficial. What is the origin of the ESM biases according to reconstruction at least for the main ones? Has this paper filled the initial objectives?*

**Response:** Thank you for your comment, we have revised the conclusion.

**Revised text:**

[revised manuscript text omitted]

**Response to comments of Anonymous Referee #2**

**1. General comments**

**Reviewer comment: (1)** *The paper of Li et al. uses a global set of pollen data to reconstruct megabiomes since the last ice age. This reconstruction is then compared with biomized outputs from Earth System Models in order to evaluate the fidelity of the model simulations. The biome reconstructions are quite interesting, and I appreciate the large fossil datasets compiled by the authors, this is huge effort to put together.*

**Response:** We appreciate the positive feedback on our work and the recognition of the effort involved in compiling the global pollen dataset for megabiome reconstructions. While we included a model-data comparison as an example study, the primary aim of our research is to present a new biomization dataset that offers the possibility to evaluate Earth System Models (ESMs). We have clarified this in our revised section "Summary and Conclusions." (Please see the next response)

**2. Specific Comments**

**Reviewer comment: (1)** *However, I have some comments about the methods as well as about the scientific contribution this makes through the interpretation of the data-model comparison. In particular, I find the conclusions to be very general and technical, and I hope that the authors are able to make the impacts of their study clearer.*

**Response:** Thank you for your comment, we have revised the Conclusions.

**Revised text:** (in red)

**Line 463-472:** "This study presents a global megabiome reconstruction for 43 timeslices at 500-year intervals over the past 21,000 years, based on the most extensive taxonomically and temporally standardized fossil pollen dataset. The dataset's reliability is supported by a high agreement (~80%) with modern potential natural vegetation, and its general consistency with the simulated paleosimulation ensemble further underscores its robustness for exploring past biome dynamics. With its high temporal and spatial coverage, it offers an unprecedented resource, not only for exploring long-term vegetation dynamics and their drivers, but also for diverse research contexts, including paleoclimate, biodiversity, and land-use studies. Furthermore, the dataset supports the evaluation of ESM-based paleo-megabiome simulations and offers insights for identifying potential biases in climate and

vegetation models. Its consistent structure and broad applicability allow us to advance our integrative understanding of past, present, and future Earth system dynamics.**"**

**Reviewer comment: (2)** *In my comment above, I say that I find the interpretation of the data-model comparison to be overly general. This is already observed in the abstract. The abstract ends with the statement: To some extent, these mismatches could be attributed to systematic model biases in the simulated climate, as well as to the different plant representations and low taxonomic resolution of pollen in the reconstructions.*

*I find this to be so general that it makes it really hard for the reader to glean any nuance the help us understand specific insights about model bias or data issues.*

*I suggest that the authors place more emphasis on the actual biome reconstruction from the pollen (which takes up more of the discussion, but is not much emphasized in the abstract) and take look deeper into the sources of mismatch to leave the reader with some key takeaways that relate directly to their stated goal (goal in abstract: to evaluate the paleosimulations from ESMs).*

**Response:** We have revised the Abstract based on your comments as follows:

**Revised text:**

**Line 23-47: "**We present a global megabiome reconstruction for 43 timeslices at 500-year intervals throughout the last 21,000 years,based on an updated and thus currently most extensive global taxonomically and temporally standardized fossil pollen dataset of 3,455 records. The evaluation with modern potential natural vegetation distributions yields an agreement of ~80%, suggesting a high reliability of the pollen-based megabiome reconstruction.

We compare the reconstruction with an ensemble of six biomized simulations derived from transient Earth System Models (ESMs). Overall, the global spatiotemporal patterns of megabiomes estimated by both, the simulation ensemble and reconstructions, are generally consistent. Specifically, they reveal a global shift from open glacial non-forest megabiomes to Holocene forest megabiomes since the Last Glacial Maximum (LGM), in line with the general climate warming trend and continental ice-sheet retreat. The shift to a global megabiome distribution generally similar to today's took place during the early Holocene; Furthermore, the reconstructions reveal that enhanced anthropogenic disturbances since the Late Holocene have not altered broad-scale megabiome patterns.

However, certain data-model deviations are evident in specific regions and periods, which could be attributed to systematic climate biases in ESMs or biases in the pollen-based biomization method. For example, at a global scale over the last 21,000 years, the largest deviations between reconstructions and simulation ensemble are observed during the LGM and early deglaciation periods. These discrepancies are probably attributed to the ESM systematic summer cold biases that overestimate tundra in periglacial regions, as well as the challenging identification of steppes and tundra from the Tibetan Plateau pollen records. Moderate deviations during the Holocene mainly occur in non-forest megabiomes in the Mediterranean and North Africa, with increasing discrepancies over time. These deviations may result from the underestimation of woody PFT cover in simulations

due to systematic biases, such as overly warm summers with dry winters in the Mediterranean, and the overrepresentation of woody taxa in reconstructions, misclassifying deserts as savanna in North Africa.

Overall, our reconstruction, with its relatively high temporal and spatial resolution, serves as a robust dataset for evaluating ESM-based paleo-megabiome simulations, as well as providing potential clues for improving systematic model biases."

**Reviewer comment: (3)** *This continues in the discussion with some very general statements about the sources of uncertainty in the comparison such as: We assume that the simulations used in this study share this rather common problem of a cold bias in boreal latitudes, resulting in the overestimation of tundra in simulations. There is no reference here to support the assumption and no sign that this potential bias was evaluated for the simulations used in the study.*

**Response:** We do not have pollen-independent climate reconstructions. So we cannot really say when and why the models have a climate that is too different, causing differences in vegetation compared to the reconstructions. There are some other reconstructions available, but it is not always clear which proxy records which climate variable, and deciding this is beyond our expertise, and a fair comparison of the past model climate with the reconstructions would be a huge effort and beyond the scope of this paper. Consequently, we limited the evaluation of the simulated climate to a comparison of the 0ka BP time-slice with modern observed climate data, which we added in the revised text.

**New text in Method:**

**Line 185-199: "**Modern observational climate data provide a crucial foundation for the assessment of climate simulations. The Climatic Research Unit gridded Time Series (CRU TS hereafter), version 4.08, is a widely used modern observational climate dataset covering all land domains of the world except Antarctica (spatial resolution: ~0.5°x0.5° on a Gaussian grid; Harris et al., 2020). The CRU TS dataset is interpolated from extensive networks of weather station observations and provides monthly temperature and precipitation data from 1901-2023 C.E. However, the early records (i.e., < 1930 C.E.) of this dataset may have high uncertainty due to sparser observation networks (Duan et al., 2024), and the late records (i.e., > 1970 C.E.) is strongly influenced by anthropogenic $CO_2$ increases (Cheng et al., 2022). We, therefore, selected monthly climatological means from 1931-1970 to generate more biologically meaningful bioclimatic variables for evaluating climate simulations at 0 cal. ka BP (O'Donnell and Ignizio, 2012; Supplementary Data 7). These bioclimatic variables represent extreme or limiting environmental factors, namely, mean temperature of warmest quarter ($T_{warm}$), mean temperature of coldest quarter ($T_{cold}$), precipitation of warmest quarter ($P_{warm}$), and precipitation of coldest quarter ($P_{cold}$). Temperature is given in degrees Celsius (°C), precipitation in millimeters (mm), and a quarter is a period of three consecutive months (1/4 of the year).**"**

**Revised text in Results and Discussion:**

**Line 185-199: "**The agreement between modern potential megabiomes and simulated megabiomes at timeslice 0 cal. ka BP is higher for the ESM-representative megabiome (cf. Sect. 2.3) than for individual ESM-based simulations (64.1% vs. 20.0–60.2%; Table 3). As a result, the ESM-representative megabiome depicts more

reliable patterns of megabiome dynamics and distribution than individual simulations, with higher agreement especially in Alaska, the Iberian Peninsula, the Alps, the Atlantic Coastal Plain of North America, and the southeastern United States (Fig. 1 and Fig. A2). However, there are still certain regions with low agreement, probably due to climatic biases. These include nearly all highlands (such as the central-southern Rocky Mountains, the central Andes, and the Tibetan Plateau) for which an overestimation of the temperature can be expected in the models due to a much lower orography than in reality caused by the smoothing in the coarse spatial resolution (3.75°x3.75° and 5°x5°) of the model grids (Fig. A3a–b). All models simulate non-forest megabiomes instead of forest in the Mediterranean region, which can be attributed to the models simulating a climate that is too seasonally dry, with, for example, too-warm summers and too-dry winters (Fig. A3a, d). The TRACE-21K simulation as well as the MPI-ESM simulations fail to reproduce the boreal forest (BOFO) in Alaska, which is then also reflected in the ESM-representative megabiomes. This failure is likely due to the simulated climate being too cold in this region, preventing the establishment of boreal forests under modeled conditions (Fig. A3a, d). Similar to the reconstructions, the transition zones between temperate forest (TEFO) and non-forest megabiomes, such as the East Asian summer monsoon margin, are regions with lower simulated megabiome agreement to the modern potential megabiome distribution. In North Africa, the models also tend to underestimate the northern extension of the grassland and dry shrubland (STEP) and incorrectly assign (warm) savanna and dry woodland (SAVA) records to tropical forest (TRFO). This is related to the biomization procedure for the model results that only relies on simulated vegetation cover fractions and simulated climate, whereas savannas are additionally determined by other ecological processes such as fire intensity and frequency (Dallmeyer et al., 2019) or grazing (van Langevelde et al., 2019)."

**Line 429-439:** "Different estimates of tundra in the circum-Arctic areas and the Tibetan Plateau are the primary sources of the strong global data-model deviations during the LGM and early deglaciation periods (Fig. 4d) at 21 and 16 cal. ka BP (Fig. 3). We observe inconsistent estimates of tundra (TUND) and boreal forest (BOFO) from the pollen-based reconstructions and the ESM-based simulations in northern Siberia (AS1), Alaska (NA1), and the East Siberian Highlands (AS2). To some extent, this mismatch could be attributed to systematic model biases in the simulated climate, as climate models tend to underestimate summer temperature in the periglacial areas compared to proxy-based reconstructions, as previously indicated in studies with different models (Deplazes et al., 2013; Alley, 2000) for that period. The simulations used in this study, especially the MPI-ESM and TRACE-21K simulations, also share this rather common problem in modern times, i.e. a summer cold bias in boreal latitudes (Fig. A3a and Table A1), resulting in an overestimation of tundra in the simulations. However, CLIMBER-X simulations perform better in these regions because they overestimate summer temperatures and produce more boreal forests."

**New figure (Appendix A3):**

[Figure]

[Figure]

**Figure A3. Differences in bioclimatic variables between ESM-based simulations at 0 cal. ka BP and observations.** The bioclimatic variables include **(a)** mean temperature of warmest quarter ($T_{warm}$), **(b)** mean temperature of coldest quarter ($T_{cold}$), **(c)** precipitation of warmest quarter ($P_{warm}$), and **(d)** precipitation of coldest quarter ($P_{cold}$). The spatial resolutions are 3.75 degrees for the MPI-ESM and TRACE-21K models and 5 degrees for the CLIMBER-X model. Note that the map legend shows bioclimatic variable values from the 5th to 95th percentile, with values above the 95th percentile shown in the 95th percentile color and values below the 5th percentile in the 5th percentile color. Notable biases include overestimated temperatures ($T_{warm \, and} \, T_{cold}$) in highlands (e.g., Rockies, Andes, Tibetan Plateau), excessively dry Mediterranean summers ($P_{warm}$), and colder-than-observed conditions in Alaska ($T_{warm \, and} \, T_{cold}$).

**Reviewer comment: (4)** *Another point I would add here is that the authors acknowledge the limitation of their modern validation exercise in incorporating human land use impacts to ecosystems when comparing with models that don't include such impacts. Please expand on how this might also impact data-model comparisons of paleo-simulations.*

**Response:** Thank you for your comment. We acknowledge the limitation of our modern validation exercise in not fully accounting for human land-use impacts, as the ESMs used in our comparisons do not include anthropogenic modifications. This has been clarified in the revised text, noting that while localized human activities may influence vegetation patterns, their impact on broad spatial and long-term paleo-simulations appears limited.

**Revised text:**

**Line 290-439:** "(c) Anthropogenic modification of pollen assemblages has, to some extent, contributed to mismatches in forested areas. For example, incorrectly reconstructed grasslands and dry shrublands (STEP) in Northern China may reflect intensive land use (e.g., deforestation)."

**Line 290-439:** "Mid-Holocene to Late Holocene (represented by the timeslices 6 and 3 cal. ka BP): The spatial patterns of megabiome distributions during this period are only slightly different from those of the early Holocene. TRFO, for example, expanded in Mesoamerican reconstructions and simulations. It is also worth noting that the forest megabiomes have not obviously shifted since the Late Holocene, as revealed by both reconstructions and simulations. Given that the simulated vegetation was in a quasi-equilibrium with the climate and unaffected by humans, this implies a relatively stable climate in that period. Therefore, we propose that enhanced anthropogenic disturbances over this time period did not promote forest degradation at a broad spatial scale, and that biomization is robust regarding these disturbance (Prentice et al., 1996; Gotanda et al., 2008)."

**Reviewer comment: (5)** *What about the impact of fire, I assume this is not included in ESMs? Could the lack of these processes in the models result in mismatch?*

**Response:** Thank you for your comment. All models used in this study include a fire module, which has been clarified in the revised text:

**Revised text:**

**Line 167-172:** "The dynamic vegetation in all models is represented by different sets of plant functional types (PFTs) that can coexist in the grid-cells. The occurrence of each PFT is constrained by fixed temperature thresholds, and the dynamics of PFT cover fraction depends for instance on the moisture availability and plant requirements. Disturbances such as fire, which are already coupled in the dynamic vegetation modules, regularly reduce the coverage of tree and shrub PFTs while promoting the expansion of herbaceous PFTs (Burton et al., 2019; Reick et al., 2021; Dallmeyer et al., 2022)."

**3. Minor comments**

**Reviewer comment: (1)** *Abstract: line 31 I don't understand term: global spatial megabiome. Does this mean the megabiomes of any particular time slice? I think there should be a way to simplify this.*

**Response:** We have revised the text as follows:

**Revised text:** (in red)

**Line 32-33:** "The shift to a global megabiome distribution generally similar to today's took place during the early Holocene."

**Reviewer comment: (2)** *line 79: "8 of our own new records" there are no references for these.*

**Response:** We have included an overview table in Supplementary Data 1, which provides site metadata and lists the references for all records.

**Revised text:**

**Line 82-85:** "Also, 52 records from the Abrupt Climate change and Environmental Responses (ACER) 1.0 database (Sánchez Goñi et al., 2017a,b), 177 records from the Chinese fossil pollen dataset (Cao et al., 2022; Zhou et al., 2023), and 8 of our own new records (AWI; for a detailed description see Supplementary Data 1) were included."

**Reviewer comment: (3)** *line 71: 3691 pollen records are in this compilation, but how many are included in the analysis after data filtration and quality control? This question applies to the numbers of records in Table 1 as well.*

**Response:** We have revised the text and updated Table 1 as follows:

**Revised text:**

**Line 82-85:** "We converted pollen data from LegacyPollen 2.0 into megabiomes using the biomization method of Prentice et al. (1996). We only analyzed records over the last 21,000 years, resulting in a final megabiome dataset of 55,868 samples at 500-year intervals from 3,455 records (Supplementary Data 1 and Data 4)."

**Revised table:**

**Table 1:** Overview of the number of pollen records, pollen taxa, plant functional types (PFTs), and megabiomes used in the biomization procedures, along with references to biomization schemes by continent. The lists of taxa-PFTs and PFTs-megabiome assignments are available in Supplementary Data 2 and Data 3.

| Continent | Pollen records | Taxa | PFTs | Megabiomes | References |
|---|---|---|---|---|---|
| Europe | 1,359 | 243 | 41 | 7 | Ni et al. (2014) |
| | | | | | Binney et al. (2017) |
| | | | | | Marinova et al. (2018) |
| | | | | | Cao et al. (2019) |
| Asia | 636 | 424 | 49 | 8 | Chen et al. (2010) |
| | | | | | Ni et al. (2014) |
| | | | | | Binney et al. (2017) |
| | | | | | Tian et al. (2018) |
| | | | | | Cao et al. (2019) |
| North America | 1,078 | 393 | 47 | 8 | Thompson and Anderson (2000) |
| | | | | | Ortega-Rosas et al. (2008) |
| | | | | | Bigelow et al. (2003) |
| | | | | | Ni et al. (2014) |
| | | | | | Cao et al. (2019) |
| Africa | 145 | 556 | 8 | 6 | Vincens et al. (2006) |
| | | | | | Lézine et al. (2009) |
| Indo-Pacific | 60 | 429 | 22 | 8 | Pickett et al. (2004) |
| South America | 177 | 576 | 19 | 8 | Marchant et al. (2001 & 2009) |
| **Total** | 3,455 | 1,447 | 98 | 8 | |

**Reviewer comment: (4)** *line 82-83: following the previous question, I know this paper which provides recommendations for data best practices, but it doesn't specify specific practices for any study. Could you please tell us specifically how you filtered data? How many dates were required for age models, how were age models generated, how many pollen samples or counts were requires, etc?*

**Response:** We have revised the text according to your suggestion as follows:

**Revised text:**

**Line 89-96:** "To improve comparability between pollen records as well as data quality, we followed the practices recommended by Flantua et al. (2023) for large-scale paleoecological data synthesis when updating the dataset. Specifically, the following key steps were involved: first, metadata of pollen records from different data sources

were examined to avoid duplicate inclusion; second, age-depth models were re-estimated for each record ($\geq 2$ radiocarbon dates) using Bacon (Blaauw and Christen, 2011; for a detailed description, see Li et al., 2022); third, pollen morphotypes were harmonized to reduce the effect of taxonomic uncertainty and nomenclatural complexity, i.e. woody taxa and major herbaceous taxa have been harmonized to genus level and other herbaceous taxa to family level (for a detailed description see Herzschuh et al., 2022)."

**Reviewer comment: (5)** *Line 90: great that dois for specific datasets were included! This is great, helps attribute credit to individual record generators!*

**Response:** We appreciate your recognition of the inclusion of DOIs for specific datasets.

**Reviewer comment: (6)** *Line 110-111 "Larix and Pinus were multiplied by factors of 15 and 0.5" This is probably sensible for NA and Europe, but what about other overrepresented taxa in other regions?*

**Response:** We now acknowledge that the lack of calibration for taxa other than Larix and Pinus somewhat limits the accuracy of reconstructions in certain regions. However, we consider our global-scale, pollen-based megabiome reconstructions reliable because record-by-record comparisons of reconstructed megabiomes at timeslice 0 cal. ka BP from 2,232 available records with modern potential megabiomes indicate an 80.2% agreement.

**Revised text:**

**Line 269-275:** "(a) The different pollen representation (including production, dispersion, and preservation) of plant taxa is the principal reason for inadequate separation of forest and open landscape ecotones. For example, the high pollen productivity of key taxa (such as *Artemisia*; Xu et al., 2014) results in an overestimation of grasslands and dry shrublands (STEP) in the East Asian summer monsoon northern marginal zone and the Great Plains of North America. Studies on pollen productivity and dispersal ability to date are mostly limited to a few taxa in north-central Europe and China (Wieczorek and Herzschuh, 2020), which limits large-scale calibration of pollen representation."

**Reviewer comment: (7)** *Line 121: "Furthermore, the assignment of pollen taxa to megabiomes and biomization routines were performed independently for each continent." Is there specific information on the differences for each continent in that supplementary material or somewhere else? This is not clear to me. For example, different harmonization schemes have been published for different geographic areas, but I don't see a reference to this or other geographically specific procedures.*

**Response:** Yes, we have included the taxa-to-PFT-to-megabiome assignment scheme for each continent in the supplementary materials. We have clarified this point in the revised text:

**Revised text:**

**Line 129-130:** "Furthermore, the assignment of pollen taxa to megabiomes and biomization routines were performed independently for each continent (Table 1; Supplementary Data 2 and Data 3)."

**Reviewer comment: (8)** *Line 159: regarding the tool of Dallmeyer et al., 2021, please provide a few details about how this works.*

**Response:** We have added details about how the tool of Dallmeyer et al., 2021 works. Please refer to the revised text below:

**Revised text:**

**Line 129-130:** "The PFTs distributions are converted into the same eight megabiomes used in the reconstructions by applying the tool of Dallmeyer et al. (2019). This tool converts the simulated PFT distributions based on assumptions of the minimum PFT cover fractions that are needed for the assignment of steppe/tundra or forest biomes and bioclimatic constraints derived from 2 m surface temperature distributions to distinguish different forest biomes (for a detailed description see Dallmeyer et al., 2019). These constraints largely adhere to the limitation rules used in the classical biome models such as BIOME4 (Kaplan et al., 2003)."

**References:**

[revised manuscript text omitted]

---

## Author Response (AR2)

**Global biome changes over the last 21,000 years inferred from model-data comparisons**

**Response to comments of editor**

**Reviewer comment: (1)** *Further clarification on the role of human land use and fire in model-data mismatches would enhance the discussion.*

**Response:** Thank you for your comment. The simulation we used included fire but not land use, which was stated in the text, so we have clarified the role of land use in the data-model comparison in the revised text.

**Text: Line 169-174: "**The dynamic vegetation in all models is represented by different sets of plant functional types (PFTs) that can coexist in the grid-cells. The occurrence of each PFT is constrained by fixed temperature thresholds, and the dynamics of PFT cover fraction depends for instance on the moisture availability and plant requirements. Disturbances such as fire, which are already coupled in the dynamic vegetation modules, regularly reduce the coverage of tree and shrub PFTs while promoting the expansion of herbaceous PFTs (Burton et al., 2019; Reick et al., 2021; Dallmeyer et al., 2022). Land use is not included in any of these simulations.**"**

**Revised text:** (in red)

**Line 464-481:** "Different estimates of non-forest megabiomes in relatively semi-arid zones, such as North Africa and the Mediterranean, have contribute to moderate but increasing data-model deviations since the early deglaciation (Fig. 4e). As shown in Fig. 3, with the transition from the glacial to the Holocene, the Mediterranean-Black Sea-Caspian Corridor (EU2) and the Mediterranean coast of northern Africa have gradually been dominated by temperate forests (TEFO) in the reconstructions, rather than grasslands and dry shrublands (STEP) in the simulations. Since the reconstructions better reproduces the region's modern potential natural vegetation than the simulation (Table 3), we infer that the simulations likely underestimated the cover fraction of woody PFTs in the simulations throughout the Holocene. Given that anthropogenic disturbances (e.g., land use and deforestation) did not promote large-scale forest degradation in this region (cf. Sect. 3.2), this underrepresentation could be attributed to the systematic model biases of hotter summers and drier winters (García-Herrera and Barriopedro, 2018; Fig. A3a–b). In addition, data-model deviations in the Sahara (AF1) are primarily observed during the Holocene, resulting from a mismatch between simulated deserts (DESE) and reconstructed savanna (SAVA). In the simulations, the weakening of the North African monsoon system led to desert expansion in response to seasonal insolation changes, a pattern supported by both proxy-based reconstructions (deMenocal et al., 2000; Shanahan et al., 2015) and climate simulations (Dallmeyer et al., 2021). However, in our reconstructions, the overrepresentation of woody taxa (e.g., *Acacia* and Arecaceae) resulted in the classification of some desert regions as savanna and dry woodlands (SAVA), potentially contributing to the increasing data-model deviations in the Sahara during the Holocene."

**Reviewer comment: (2)** *The authors should ensure a precise explanation of biome interpolation in ESM grids.*

**Response:** Thank you for your comment and we have revised the text.

**Revised text:** (in red)

**Line 244-250:** "We aggregated the records into regular longitude-latitude grid-cells of size 3.75°x3.75° to reduce the sampling bias from the non-uniform spatial distribution of records and to facilitate a more direct model-data comparison. At each timeslice, the reconstructed or simulated megabiome assigned to a grid-cell was determined based on the most frequently occurring megabiome among the available records in that grid-cell. When multiple megabiomes had the same highest frequency, we applied the same criterion used in pollen-based reconstructions, prioritizing the highest-frequency megabiome with the fewest PFTs and taxa. Similarly, the data-model EMDs for each grid-cell were derived as the median EMDs of the available records within that grid-cell."

**Reviewer comment: (3)** *Minor technical clarifications, such as references for newly included pollen records, should be incorporated.*

**Response:** We newly added the description of the archived dataset in the revised text.

**Revised text:** (in red)

**Line 97-107:** "The LegacyPollen 2.0 dataset is archived in the PANGAEA repository (https://doi.org/10.1594/PANGAEA.965907; Li et al., 2025) and is open-access. It follows the framework of LegacyPollen 1.0 dataset (Herzschuh et al., 2021, 2022), providing pollen count and pollen percentage data per continent, a taxa harmonization master table, and site metadata (such as data sources, Dataset ID, site name, location, archive type, site description, and references). To enhance data traceability and ensure high-quality standards, we have newly incorporated the Neotoma digital object identifier (DOI) into the metadata for Neotoma-derived records, allowing direct linkage to the living Neotoma database and reducing the risk of data staleness. These DOIs were generated with the *doi*() function from the package *neotoma2* (version 1.0.3; Socorro and Goring, 2024) in the R software environment (version 4.4.1; R Core Team, 2023). Additionally, we also newly added the PANGAEA Event (PANGAEA dataset identifier) for each record to ensure that our dataset meets PANGAEA's high standards for quality, usability, and compliance."